# BOIL: Towards Representation Change for Few-shot Learning

**Jaehoon Oh**[*1]**, Hyungjun Yoo**[*1]**, ChangHwan Kim**[1] **& Se-Young Yun**[2]
[1]Graduate School of Knowledge Service Engineering, KAIST
[2]Graduate School of Artificial Intelligence, KAIST
`{jaehoon.oh,yoohjun,kimbob,yunseyoung}@kaist.ac.kr`

## Abstract

Model Agnostic Meta-Learning (MAML) is one of the most representative of gradient-based meta-learning algorithms. MAML learns new tasks with a few data samples using inner updates from a meta-initialization point and learns the meta-initialization parameters with outer updates. It has recently been hypothesized that *representation reuse*, which makes little change in efficient representations, is the dominant factor in the performance of the meta-initialized model through MAML in contrast to *representation change*, which causes a significant change in representations. In this study, we investigate the necessity of *representation change* for the ultimate goal of few-shot learning, which is solving domain-agnostic tasks. To this aim, we propose a novel meta-learning algorithm, called *BOIL* (Body Only update in Inner Loop), which updates only the body (extractor) of the model and freezes the head (classifier) during inner loop updates. BOIL leverages *representation change* rather than *representation reuse*. This is because feature vectors (representations) have to move quickly to their corresponding frozen head vectors. We visualize this property using cosine similarity, CKA, and empirical results without the head. BOIL empirically shows significant performance improvement over MAML, particularly on cross-domain tasks. The results imply that *representation change* in gradient-based meta-learning approaches is a critical component.

## 1 Introduction

Meta-learning, also known as "learning to learn," is a methodology that imitates human intelligence that can adapt quickly with even a small amount of previously unseen data through the use of previous learning experiences. To this aim, meta-learning with deep neural networks has mainly been studied using metric- and gradient-based approaches. Metric-based meta-learning (Koch, 2015; Vinyals et al., 2016; Snell et al., 2017; Sung et al., 2018) compares the distance between feature embeddings using models as a mapping function of data into an embedding space, whereas gradient-based meta-learning (Ravi & Larochelle, 2016; Finn et al., 2017; Nichol et al., 2018) quickly learns the parameters to be optimized when the models encounter new tasks.

Model-agnostic meta-learning (MAML) (Finn et al., 2017) is the most representative gradient-based meta-learning algorithm. MAML algorithm consists of two optimization loops: an inner loop and an outer loop. The inner loop learns task-specific knowledge, and the outer loop finds a universally good meta-initialized parameter allowing the inner loop to quickly learn any task from the initial point with only a few examples. This algorithm has been highly influential in the field of meta-learning, and numerous follow-up studies have been conducted (Oreshkin et al., 2018; Rusu et al., 2018; Zintgraf et al., 2018; Yoon et al., 2018; Finn et al., 2018; Triantafillou et al., 2019; Sun et al., 2019; Na et al., 2019; Tseng et al., 2020).

Very recent studies (Raghu et al., 2020; Arnold et al., 2019) have attributed the success of MAML to high-quality features before the inner updates from the meta-initialized parameters. For instance, Raghu et al. (2020) claimed that MAML learns new tasks by updating the head (the last fully connected layer) with almost the same features (the output of the penultimate layer) from the meta-initialized network. In this paper, we categorize the learning patterns as follows: A small change in the representations during task learning is named *representation reuse*, whereas a large change is named *representation change*.[1] Thus, *representation reuse* was the common belief of MAML.

---

[*]The authors contribute equally to this paper.

[1]In our paper, *representation reuse* and *representation change* correspond to *feature reuse* and *rapid learning* in (Raghu et al., 2020), respectively. To prevent confusion from terminology, we re-express the terms.

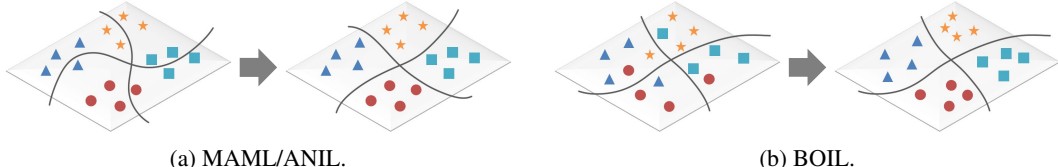

(a) MAML/ANIL.                                         (b) BOIL.

Figure 1: **Difference in task-specific (inner) updates between MAML/ANIL and BOIL**. In the figure, the lines represent the decision boundaries defined by the head (classifier) of the network. Different shapes and colors mean different classes. (a) MAML mainly updates the head with a negligible change in body (extractor); hence, representations on the feature space are almost identical. ANIL does not change in the body during inner updates, and they are therefore identical. However, (b) BOIL updates only the body without changing the head during inner updates; hence, representations on the feature space change significantly with the fixed decision boundaries. We visualize the representations from various data sets using UMAP (Uniform Manifold Approximation and Projection for dimension reduction) (McInnes et al., 2018) in Appendix B.

Herein, we pose an intriguing question: *Is representation reuse sufficient for meta-learning?* We believe that the key to successful meta-learning is closer to *representation change* than to *representation reuse*. More importantly, *representation change* is crucial for cross-domain adaptation, which is considered the ultimate goal of meta-learning. By contrast, the MAML accomplished with *representation reuse* might be poorly trained for cross-domain adaptation since the success of *representation reuse* might rely heavily on the similarity between the source and the target domains.

To answer this question, we propose a novel meta-learning algorithm that leverages *representation change*. Our contributions can be summarized as follows:

- We emphasize the necessity of *representation change* for meta-learning through cross-domain adaptation experiments.

- We propose a simple but effective meta-learning algorithm that *learns the Body (extractor) of the model Only in the Inner Loop* (BOIL). We empirically show that BOIL improves the performance over most of benchmark data sets and that this improvement is particularly noticeable in fine-grained data sets or cross-domain adaptation.

- We interpret the connection between BOIL and the algorithm using preconditioning gradients (Flennerhag et al., 2020) and show their compatibility, improving performance.

- We demonstrate that the BOIL algorithm enjoys *representation layer reuse* on the low-/mid-level body and *representation layer change* on the high-level body using the cosine similarity and the Centered Kernel Alignment (CKA). We visualize the features between before and after an adaptation, and empirically analyze the effectiveness of the body of BOIL through an ablation study on eliminating the head.

- For ResNet architectures, we propose a disconnection trick that removes the back-propagation path of the last skip connection. The disconnection trick strengthens *representation layer change* on the high-level body.

## 2    PROBLEM SETTING

### 2.1    META-LEARNING FRAMEWORK (MAML)

The MAML algorithm (Finn et al., 2017) attempts to meta-learn the best initialization of the parameters for a task-learner. It consists of two main optimization loops: an inner loop and an outer loop. First, we sample a batch of tasks within a data set distribution. Each task $\tau_i$ consists of a support set $S_{\tau_i}$ and a query set $Q_{\tau_i}$. When we sample a support set for each task, we first sample $n$ labels from the label set and then sample $k$ instances for each label. Thus, each support set contains $n \times k$ instances. For a query set, we sample instances from the same labels with the support set.

With these tasks, the MAML algorithm conducts both meta-training and meta-testing. During meta-training, we first sample a meta-batch consisting of $B$ tasks from the meta-training data set. In the

*inner loops*, we update the meta-initialized parameters $\theta$ to task-specific parameters $\theta_{\tau_i}$ using the *task-specific* loss $L_{S_{\tau_i}}(f_\theta)$, where $f_\theta$ is a neural network parameterized by $\theta$, as follows:[2]

$$\theta_{\tau_i} = \theta - \alpha \nabla_\theta L_{S_{\tau_i}}(f_\theta) \tag{1}$$

Using the query set of the corresponding task, we compute the loss $L_{Q_{\tau_i}}(f_{\theta_{\tau_i}})$ based on each inner updated parameter. By summing all these losses, the *meta-loss* of each meta-batch, $L_{meta}(\theta)$, is computed. The meta-initialized parameters are then updated using the meta-loss in the *outer loop* through a gradient descent.

$$\theta' = \theta - \beta \nabla_\theta L_{meta}(\theta), \text{ where } L_{meta}(\theta) = \sum_{i=1}^{B} L_{Q_{\tau_i}}(f_{\theta_{\tau_i}}) \tag{2}$$

In meta-testing, the inner loop, which can be interpreted as task-specific learning, is the same as in meta-training. However, the outer loop only computes the accuracy using a query set of tasks and does not perform a gradient descent; thus, it does not update the meta-initialization parameters.

## 2.2 EXPERIMENTAL SETUP

We used two backbone networks, **4conv network** with 64 channels from Vinyals et al. (2016) and **ResNet-12** starting with 64 channels and doubling them after every block from Oreshkin et al. (2018). For the batch normalization, we used batch statistics instead of the running statistics during meta-testing, following the original MAML (Finn et al., 2017). We trained 4conv network and ResNet-12 for 30,000 and 10,000 epochs, respectively, and then used *the model with the best accuracy on meta-validation data set* to verify the performance. We applied an inner update *once* for both meta-training and meta-testing. The outer learning rate was set to 0.001 and 0.0006 and the inner learning rate was set to 0.5 and 0.3 for 4conv network and ResNet-12, respectively. All results were reproduced by our group and reported as the average and standard deviation of the accuracies over $5 \times 1,000$ tasks, and the values in parentheses in the algorithm name column of the tables are the number of shots. We validated both MAML/ANIL and BOIL on two general data sets, **miniImageNet** (Vinyals et al., 2016) and **tieredImageNet** (Ren et al., 2018), and two specific data sets, **Cars** (Krause et al., 2013) and **CUB** (Welinder et al., 2010). Note that our algorithm is not for state-of-the-art performance but for a proposal of a new learning scheme for meta-learning. Full details on the implementation and data sets are described in Appendix A.[3] In addition, the results of the other data sets at a size of $32 \times 32$ and using the 4conv network with 32 channels from Finn et al. (2017) (i.e., original setting) are reported in Appendix C and Appendix D, respectively.

## 3 BOIL (BODY ONLY UPDATE IN INNER LOOP)

### 3.1 THE ULTIMATE GOAL OF META-LEARNING: DOMAIN-AGNOSTIC ADAPTATION

Recently, Raghu et al. (2020) proposed two opposing hypotheses, *representation reuse* and *representation change*, and demonstrated that *representation reuse* is the dominant factor of MAML. We can discriminate two hypotheses according to which part of the neural network, body or head, is mostly updated through the inner loop. Here, the body indicates all convolutional layers, and the head indicates the remaining fully connected layer. In other words, the *representation change* hypothesis attributes the capability of MAML to the updates on the body, whereas the *representation reuse* hypothesis considers that the network body is already universal to various tasks before the inner loops. To demonstrate the *representation reuse* hypothesis of MAML, the authors proposed the ANIL (Almost No Inner Loop) algorithm, which only updates the head in the inner loops during training and testing, and showed that ANIL has a performance comparable to that of MAML. This implies that the representation trained by MAML/ANIL, even before updated task-specifically, is sufficient

---

[2]Although the inner loop(s) can be applied through one or more steps, for simplicity, we consider only the case of a single inner loop.

[3]All implementations are based on Torchmeta (Deleu et al., 2019) except for WarpGrad, and all results were reproduced according to our details. These results are not the highest for MAML/ANIL because our setting is more fitted to BOIL. However, under more suitable hyperparameters for each algorithm, the best performance of BOIL is better than that of MAML/ANIL.

to achieve the desired performance. Furthermore, they proposed the NIL-testing (No Inner Loop) algorithm, which removes the head and performs unseen tasks using only the distance between the representations of a support set and those of a query set during testing to identify the capability of *representation reuse*. NIL-testing of MAML also achieves a performance comparable to MAML. Based on these results, it was claimed that the success of MAML is attributed to *representation reuse*.

Here, we investigate the necessity of *representation change*. We believe that the meta-trained models should achieve a good performance in many other domains, which is referred to as *domain-agnostic adaptation* in this paper. To this end, *representation reuse* is not appropriate since *representation reuse* uses the similarity between the source and target domains. The higher the similarity, the higher the efficiency. Therefore, when there are no strong similarities between the source and target domains, good representations for the source domain could be imperfect representations for the target domain. Table 2, which lists our experimental results on cross-domain tasks, shows that the MAML enjoying *representation reuse* is worse than BOIL leveraging *representation change*, which will be discussed in detail in the next section.

## 3.2 BOIL ALGORITHM

Inspired by the necessity, we design an algorithm that updates only the body of the model and freezes the head of the model during the task learning to enforce *representation change* through inner updates. Because the gradients must be back-propagated to update the body, we set the learning rate of the head to zero in the inner updates during both meta-training and meta-testing. Otherwise, the learning and evaluation procedures of BOIL are the same as those of MAML. Therefore, the computational overhead does not change.

Formally speaking, with the notations used in Section 2.1, the meta-initialized parameters $\theta$ can be separated into body parameters $\theta_b$ and head parameters $\theta_h$, that is, $\theta = \{\theta_b, \theta_h\}$. For a sample image $x \in \mathbb{R}^i$, an output can be expressed as $\hat{y} = f_\theta(x) = f_{\theta_h}(f_{\theta_b}(x)) \in \mathbb{R}^n$, where $f_{\theta_b}(x) \in \mathbb{R}^d$. The task-specific body parameters $\theta_{b,\tau_i}$ and head parameters $\theta_{h,\tau_i}$ through an inner loop given task $\tau_i$ are thus as follows:

$$\theta_{b,\tau_i} = \theta_b - \alpha_b \nabla_{\theta_b} L_{S_{\tau_i}}(f_\theta) \ \& \ \theta_{h,\tau_i} = \theta_h - \alpha_h \nabla_{\theta_h} L_{S_{\tau_i}}(f_\theta) \tag{3}$$

where $\alpha_b$ and $\alpha_h$ are the inner loop learning rates corresponding to the body and head, respectively. MAML usually sets $\alpha = \alpha_b = \alpha_h (\neq 0)$, ANIL sets $\alpha_b = 0$ and $\alpha_h \neq 0$, and BOIL sets $\alpha_b \neq 0$ and $\alpha_h = 0$.

These simple differences force the change in the dominant factor of task-specific updates, from the head to the body. Figure 1 shows the main difference in the inner updates between MAML/ANIL and BOIL. To solve new tasks, the head mainly or only changes in MAML/ANIL (Raghu et al., 2020), whereas in BOIL, the body changes.

### 3.2.1 PERFORMANCE IMPROVEMENT ON BENCHMARK DATA SETS AND CROSS-DOMAIN TASKS

Table 1: Test accuracy (%) of 4conv network on benchmark data sets. The values in parenthesis in the algorithm name column of tables are the number of shots.

| Domain | General (Coarse-grained) | | Specific (Fine-grained) | |
|---|---|---|---|---|
| Dataset | miniImageNet | tieredImageNet | Cars | CUB |
| MAML(1) | $47.44 \pm 0.23$ | $47.44 \pm 0.18$ | $45.27 \pm 0.26$ | $56.18 \pm 0.37$ |
| ANIL(1) | $47.82 \pm 0.20$ | $\mathbf{49.35} \pm 0.26$ | $46.81 \pm 0.24$ | $57.03 \pm 0.41$ |
| BOIL(1) | $\mathbf{49.61} \pm 0.16$ | $48.58 \pm 0.27$ | $\mathbf{56.82} \pm 0.21$ | $\mathbf{61.60} \pm 0.57$ |
| MAML(5) | $61.75 \pm 0.42$ | $64.70 \pm 0.14$ | $53.23 \pm 0.26$ | $69.66 \pm 0.03$ |
| ANIL(5) | $63.04 \pm 0.42$ | $65.82 \pm 0.12$ | $61.95 \pm 0.38$ | $70.93 \pm 0.28$ |
| BOIL(5) | $\mathbf{66.45} \pm 0.37$ | $\mathbf{69.37} \pm 0.12$ | $\mathbf{75.18} \pm 0.21$ | $\mathbf{75.96} \pm 0.17$ |

Table 1 and Table 2 display the superiority of BOIL on most benchmark data sets and on cross-domain adaptation tasks, where the source and target domains differ (i.e., the meta-training and meta-testing data sets are different). In Table 1, the performance improvement is particularly noticeable on the specific domain data sets Cars and CUB. The results demonstrate that *representation change* is necessary even if there is a similarity between the source and target domains. Table 2 shows that BOIL is closer to the ultimate goal of meta-learning, which is a domain-agnostic adaptation.

Table 2: Test accuracy (%) of 4conv network on cross-domain adaptation.

| adaptation | General to General | | General to Specific | | Specific to General | | Specific to Specific | |
|---|---|---|---|---|---|---|---|---|
| meta-train | tieredImageNet | miniImageNet | miniImageNet | miniImageNet | Cars | Cars | CUB | Cars |
| meta-test | miniImageNet | tieredImageNet | Cars | CUB | miniImageNet | tieredImageNet | Cars | CUB |
| MAML(1) | $47.60 \pm 0.24$ | $51.61 \pm 0.20$ | $33.57 \pm 0.14$ | $40.51 \pm 0.08$ | $26.95 \pm 0.15$ | $28.46 \pm 0.18$ | $32.22 \pm 0.30$ | $29.64 \pm 0.19$ |
| ANIL(1) | $49.67 \pm 0.31$ | $52.82 \pm 0.29$ | $34.77 \pm 0.31$ | $41.12 \pm 0.15$ | $28.67 \pm 0.17$ | $29.41 \pm 0.19$ | $33.07 \pm 0.43$ | $28.32 \pm 0.32$ |
| BOIL(1) | $\mathbf{49.74} \pm 0.26$ | $\mathbf{53.23} \pm 0.41$ | $\mathbf{36.12} \pm 0.29$ | $\mathbf{44.20} \pm 0.15$ | $\mathbf{33.71} \pm 0.13$ | $\mathbf{34.06} \pm 0.20$ | $\mathbf{35.44} \pm 0.46$ | $\mathbf{34.79} \pm 0.27$ |
| MAML(5) | $65.22 \pm 0.20$ | $65.76 \pm 0.27$ | $44.56 \pm 0.21$ | $53.09 \pm 0.16$ | $30.64 \pm 0.19$ | $32.62 \pm 0.21$ | $41.24 \pm 0.21$ | $32.18 \pm 0.13$ |
| ANIL(5) | $66.47 \pm 0.16$ | $66.52 \pm 0.28$ | $46.55 \pm 0.29$ | $55.82 \pm 0.21$ | $35.38 \pm 0.10$ | $36.94 \pm 0.10$ | $43.05 \pm 0.23$ | $37.99 \pm 0.15$ |
| BOIL(5) | $\mathbf{69.33} \pm 0.19$ | $\mathbf{69.37} \pm 0.23$ | $\mathbf{50.64} \pm 0.22$ | $\mathbf{60.92} \pm 0.11$ | $\mathbf{44.51} \pm 0.25$ | $\mathbf{46.09} \pm 0.23$ | $\mathbf{47.30} \pm 0.22$ | $\mathbf{45.91} \pm 0.28$ |

Recently, Guo et al. (2019) noted that existing meta-learning algorithms have weaknesses in terms of cross-domain adaptation. We divide the cross-domain adaptation into four cases: general to general, general to specific, specific to general, and specific to specific. Previous studies considered the cross-domain scenario from a general domain to a specific domain (Chen et al., 2019; Guo et al., 2019). In this paper, we also evaluate the reverse case. BOIL outperforms MAML/ANIL not only on the typical cross-domain adaptation scenario but also on the reverse one. In particular, the performance improvement, when the domain changes from birds (CUB as a meta-train set) to cars (Cars as a meta-test set), implies that the *representation change* in BOIL enables the model to adapt to an unseen target domain that is entirely different from the source domain.

### 3.2.2 ABLATION STUDY ON THE LEARNING RATE OF THE HEAD

In this section, we control the inner loop update learning rate of the head to verify the effect of training the head to the performance. The results are depicted in Table 3. The best performance is achieved when the learning rate is 0 (BOIL). However, the accuracy rapidly decreases as the learning rate of the head grows. Even with $1/10\times$ head learning rate compared to other layers, the test accuracies are significantly degraded. Therefore, it is thought that freezing head is crucial.

| Head's Learning Rate ($\alpha_h$) | miniImageNet | Cars |
|---|---|---|
| 0.00 (BOIL) | $\mathbf{66.45} \pm 0.37$ | $\mathbf{75.18} \pm 0.21$ |
| 0.05 | $38.81 \pm 0.21$ | $68.67 \pm 0.21$ |
| 0.10 | $49.49 \pm 0.16$ | $68.86 \pm 0.30$ |
| 0.5 (MAML in ours) | $61.75 \pm 0.42$ | $53.23 \pm 0.26$ |

Table 3: 5-Way 5-Shot test accuracy according to the learning rate of the head.

### 3.2.3 BOIL AND PRECONDITIONING GRADIENTS

Some aspects of BOIL can be explained by preconditioning gradients (Lee & Choi, 2018; Flennerhag et al., 2020). Preconditioning gradients occur when a particular layer is shared over all tasks, warping the spaces (e.g., rotating and scaling). For instance, one might consider the frozen head of BOIL to be a warp layer of the entire body (Flennerhag et al., 2020).

Preconditioning gradients can avoid overfitting in a high-capacity model (Flennerhag et al., 2020), and such a benefit is still valid with BOIL. Indeed, many prior studies have suffered from an overfitting problem, and thus it is challenging to train the backbone network more extensively than the 4conv network with 32 filters (Finn et al., 2017). By contrast, BOIL can increase the validation accuracy with more extensive networks. The accuracy of models with 32, 64, and 128 filters continues to increase to 64.02, 66.72, and 69.23, without overfitting. In Appendix E, we report these results as well as the training and valid accuracy curves of BOIL for three different network sizes, in which the larger networks are trained well. We further hypothesize that the head is the most critical part of an overfitting problem, and BOIL can succeed in dealing with the problem by simply ignoring the head in the inner loops.

However, one essential difference between BOIL and the preconditioning gradients is whether the head is frozen. Prior studies did not freeze the last fully connected layer or used any additional fully connected layer to precondition the gradients, and hence *representation reuse* is still the major factor of their training. To the best of our knowledge, BOIL is the first approach that enforces *representation change* by freezing the head in the inner loops.

To investigate the gain from *representation change*, we adapt BOIL to WarpGrad (Flennerhag et al., 2020).[4] Four different models are tested, the architectures of which are fully described in Appendix F. Table 4 shows the test accuracy of the four models, where the BOIL-WarpGrad

| Model | Accuracy |
|---|---|
| WarpGrad w/ last warp head | $83.19 \pm 0.79$ |
| WarpGrad w/o last warp head | $83.16 \pm 0.69$ |
| BOIL-WarpGrad w/ last warp conv | $83.68 \pm 0.82$ |
| BOIL-WarpGrad w/o last warp conv | $\mathbf{84.88} \pm 0.42$ |

Table 4: Test accuracy(%) of WarpGrad and BOIL-WarpGrad over $5 \times 100$ tasks.

[4] We follow the setting in https://github.com/flennerhag/warpgrad, and the details about this implementation are in Appendix F. This task is related to a long-adaptation task.

models freeze the fully connected layer from the corresponding WarpGrad model. It is observed that BOIL-WarpGrad improves WarpGrad and BOIL-WarpGrad without the last warp conv improves BOIL-WarpGrad with the last warp conv. The latter result indicates that, to support BOIL, the last convolution layer must not be fixed but rather updated during the inner loops.

# 4 REPRESENTATION CHANGE IN BOIL

## 4.1 REPRESENTATION CHANGE BEFORE/AFTER ADAPTATION

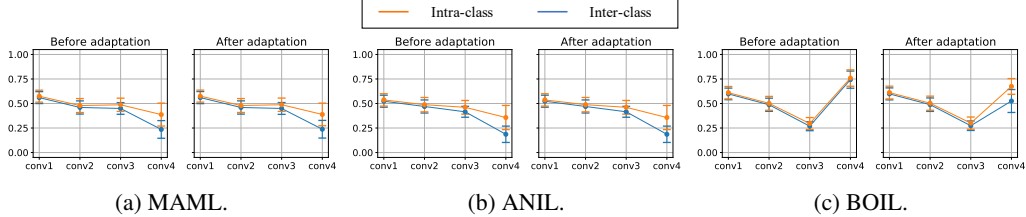

(a) MAML.                    (b) ANIL.                    (c) BOIL.

Figure 2: Cosine similarity of 4conv network.

To analyze whether the learning scheme of BOIL is *representation reuse* or *representation change*, we explore the layer-wise alteration of the representations before and after adaptation. We compute the cosine similarities and CKA values of the convolution layers with the meta-trained 4conv network (as detailed in Appendix A). We first investigate the cosine similarity between the representations of a query set including 5 classes and 15 samples per class from **miniImageNet** after every convolution module. In Figure 2, the orange line represents the average of the cosine similarity between the samples having the same class, and the blue line represents the average of the cosine similarity between the samples having different classes. In Figure 2, the left panel of each algorithm is before the inner loop adaptation, and the right panel is after inner loop adaptation.

The key observations from Figure 2, which are also discussed in Section 4.2 with other experiments, are as follows:

- The cosine similarities of MAML/ANIL (Figure 2a and Figure 2b) have the similar patterns, supporting *representation reuse*. Their patterns do not show any noticeable difference before and after adaptation. They make the average of the cosine similarities monotonically decrease and make the representations separable by classes when the representations reach the last convolution layer. These analyses indicate that the effectiveness of MAML/ANIL heavily leans on the meta-initialized body, not the task-specific adaptation.

- The cosine similarities of BOIL (Figure 2c) have a different pattern from those of MAML/ANIL, supporting *representation change*. The BOIL's pattern changes to distinguish classes after adaptation. Before adaptation, BOIL reduces the average cosine similarities only up to conv3, and all representations are concentrated regardless of their classes after the last convolution layer. Hence, BOIL's meta-initialized body cannot distinguish the classes. However, after adaptation, the similarity of the different classes rapidly decrease on conv4, which means that the body can distinguish the classes through adaptation.

- The reason why the change in BOIL before and after adaptation occurs only on conv4 is a peculiarity of the convolutional body, analyzed by Zeiler & Fergus (2014). Although the general and low-level features produced through the front convolution layers (e.g., colors, lines, and shapes) do not differ much from the task-specific adaptation, the discriminative representations produced through the last convolution layer (conv4) differ from class to class. The importance of the last convolutional layer on the performance in few-shot image classification tasks is also investigated by Arnold et al. (2019); Chen et al. (2020). These changes before and after the adaptation support the fact that BOIL enjoys *representation layer reuse* at the low- and mid-levels of the body and *representation layer change* in a high-level of the body.[5] Nevertheless, the degree of *representation layer reuse* in a low- and mid-levels in BOIL is lower than that in MAML/ANIL, which is measured using gradient norms (Appendix G). We also report the cosine similarity including head in Appendix H.

---

[5]In the (Raghu et al., 2020), *representation reuse/change* are defined at the model-level. Namely, *representation reuse* indicates that none of the representations after the convolution layers significantly change during the inner loop updates, otherwise *representation change*. We extend this concept from the model-level to the layer-level. To clarify it, we use *representation layer reuse/change* for the layer-level. Therefore, if a model has even one layer with *representation layer change*, it is said that the model follows *representation change*.

Through these observations, we believe that MAML follows the *representation reuse* training scheme, whereas BOIL follows *representation change* training scheme through *representation layer reuse* before the last convolution layer and *representation layer change* at the last convolution layer.

Next, we demonstrate that BOIL enjoys *representation layer reuse* on the low- and mid-level and *representation layer change* on the high-level of the body by computing the CKA (Kornblith et al., 2019) before and after adaptation. When the CKA between two representations is close to 1, the representations are almost identical. In Figure 3, as mentioned in Raghu et al. (2020), CKA shows that the MAML/ANIL algorithms do not change the representation in the body. However, BOIL changes the representation of the last convolution layer. This result indicates that the BOIL algorithm learns rapidly through *representation change*. In addition, the *representation change* on the **Cars** data set is described in Appendix I.

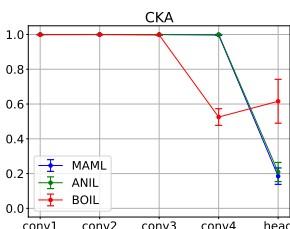

Figure 3: CKA of 4conv.

## 4.2 EMPIRICAL ANALYSIS OF REPRESENTATION CHANGE IN BOIL

Table 5: Test accuracy (%) of 4conv network according to the head's existence before/after an adaptation.

| meta-train | miniImageNet | | | | | | | |
|---|---|---|---|---|---|---|---|---|
| meta-test | miniImageNet | | | | Cars | | | |
| head | w/ head | | w/o head (NIL-testing) | | w/ head | | w/o head (NIL-testing) | |
| adaptation | before | after | before | after | before | after | before | after |
| MAML(1) | 19.96 ± 0.25 | 47.44 ± 0.23 | 48.28 ± 0.20 | 47.87 ± 0.14 | 20.05 ± 0.16 | 33.57 ± 0.14 | 34.47 ± 0.19 | 34.36 ± 0.16 |
| ANIL(1) | 20.09 ± 0.19 | 47.92 ± 0.20 | **48.86 ± 0.12** | **48.86 ± 0.12** | 20.16 ± 0.05 | 34.77 ± 0.31 | **35.48 ± 0.24** | **35.48 ± 0.24** |
| BOIL(1) | 19.94 ± 0.13 | **49.61 ± 0.16** | 24.07 ± 0.19 | 46.73 ± 0.17 | 19.94 ± 0.06 | **36.12 ± 0.29** | 23.30 ± 0.15 | 34.07 ± 0.32 |
| MAML(5) | 20.04 ± 0.17 | 61.75 ± 0.42 | 64.61 ± 0.39 | 64.47 ± 0.39 | 19.97 ± 0.18 | 44.56 ± 0.21 | 47.66 ± 0.28 | 47.53 ± 0.28 |
| ANIL(5) | 20.09 ± 0.13 | 63.04 ± 0.42 | **66.11 ± 0.51** | **66.11 ± 0.51** | 20.08 ± 0.07 | 46.55 ± 0.29 | **49.62 ± 0.20** | 49.62 ± 0.20 |
| BOIL(5) | 20.04 ± 0.21 | **66.45 ± 0.37** | 32.03 ± 0.16 | 64.61 ± 0.27 | 20.06 ± 0.16 | **50.64 ± 0.22** | 30.33 ± 0.18 | **50.40 ± 0.30** |

Table 5 describes the test accuracy on miniImageNet and Cars of the model meta-trained on miniImageNet before and after an inner update according to the presence of the head. To evaluate the performance in a case without a classifier, we first create a template of each class by averaging the representations from the support set. Then, the class of the sample from the query set is predicted as the class whose template has the highest cosine similarity with the representation of the sample. This is the same as NIL-testing in (Raghu et al., 2020).

The results provide some intriguing interpretations:

- **With the head for all algorithms.** Before adaptation, all algorithms on the same- and cross-domain are unable to distinguish all classes (20%). This status could be considered as an optimum of meta-initialization. We also discuss it in Appendix L. In BOIL, the representations have to move quickly to their corresponding frozen head. Moreover, after adaptation, BOIL overwhelms the performance of the other algorithms. This means that *representation change* of BOIL is more effective than *representation reuse* of MAML/ANIL.

- **Without the head in MAML/ANIL.** In this setting, representations from the body are evaluated before and after adaptation. Before adaptation, MAML and ANIL already generate sufficient representations to classify, and adaptation makes little or no difference. The former observation matches the cosine similarity gap between an intra- and inter-class on each left panel in Figure 2a and Figure 2b, and the latter observation matches the CKA values of close to or exactly 1 on conv4 of MAML/ANIL in Figure 3.

- **Without the head in BOIL.** BOIL shows a steep performance improvement through adaptation on the same- and cross-domain. This result implies that the body of BOIL can be task-specifically updated. It is matched with Figure 2c, where the cosine similarity gap between the intra-class and inter-class on the space after conv4 is near zero before adaptation but increases after adaptation. This implies that the poor representations can be dramatically improved in BOIL. Therefore, the low CKA value on conv4 of BOIL in Figure 3 is natural.

To summarize, the meta-initialization by MAML and ANIL provides efficient representations through the body even before adaptation. By contrast, although BOIL's meta-initialization provides less efficient representations compared to MAML and ANIL, the body can extract more efficient representations through task-specific adaptation based on *representation change*.

Note that the penultimate layer (i.e., conv4) of BOIL acts differently from the output layer (i.e., head) of MAML/ANIL. The penultimate layer of BOIL might seem like a pseudo-head layer, but not at all. The output layer of MAML/ANIL is adapted based on the well-represented features (i.e., features after conv4). In contrast, the penultimate layer of BOIL is adapted based on the poorly-represented features (i.e., features after conv3). It means that the head layer's role of MAML/ANIL is to draw a simple decision boundary for the high-level features represented by the output of the last convolutional layer. However, the penultimate layer of BOIL acts as a non-linear transformation so that the fixed output head layer can effectively conduct a classifier role. We also report empirical analyses of penultimate layer of BOIL and an ablation study on learning layer in Appendix J and Appendix K.

## 5 BOIL TO A LARGER NETWORK

Many recent studies (Vuorio et al., 2019; Rusu et al., 2018; Sun et al., 2019) have used deeper networks such as ResNet (He et al., 2016), Wide-ResNet (Zagoruyko & Komodakis, 2016), and DenseNet (Huang et al., 2017) as a backbone network. The deeper networks, in general, use feature wiring structures to facilitate the feature propagation. We explore BOIL's applicability to a deeper network with the wiring structure, ResNet-12, and propose a simple trick to boost *representation change* by disconnecting the last skip connection. This trick is described in Section 5.1.

Table 6: 5-Way 5-Shot test accuracy (%) of ResNet-12. LSC means Last Skip Connection.

| Meta-train | miniImageNet | | | Cars | | |
|---|---|---|---|---|---|---|
| Meta-test | miniImageNet | tieredImageNet | Cars | Cars | miniImageNet | CUB |
| MAML w/ LSC | $68.51 \pm 0.39$ | $71.67 \pm 0.13$ | $43.46 \pm 0.15$ | $75.49 \pm 0.20$ | $34.42 \pm 0.06$ | $35.87 \pm 0.19$ |
| MAML w/o LSC | $67.87 \pm 0.22$ | $70.31 \pm 0.10$ | $41.40 \pm 0.11$ | $73.63 \pm 0.11$ | $37.65 \pm 0.11$ | $34.77 \pm 0.26$ |
| ANIL w/ LSC | $68.54 \pm 0.34$ | $71.93 \pm 0.11$ | $45.13 \pm 0.15$ | $79.45 \pm 0.23$ | $35.03 \pm 0.07$ | $35.09 \pm 0.19$ |
| ANIL w/o LSC | $67.20 \pm 0.13$ | $69.79 \pm 0.24$ | $43.46 \pm 0.18$ | $75.32 \pm 0.15$ | $38.15 \pm 0.15$ | $36.06 \pm 0.14$ |
| BOIL w/ LSC | $70.50 \pm 0.28$ | $71.86 \pm 0.21$ | $49.69 \pm 0.17$ | $80.98 \pm 0.14$ | $45.89 \pm 0.32$ | $43.34 \pm 0.21$ |
| BOIL w/o LSC | $\mathbf{71.30} \pm 0.28$ | $\mathbf{74.12} \pm 0.30$ | $\mathbf{49.71} \pm 0.28$ | $\mathbf{83.99} \pm 0.20$ | $\mathbf{48.41} \pm 0.18$ | $\mathbf{44.23} \pm 0.18$ |

Table 6 shows the test accuracy results of ResNet-12, which is meta-trained and meta-tested with various data sets according to the fineness of the domains. This result indicates that BOIL can be applied to other general architectures by showing a better performance than MAML not only on standard benchmark data sets but also on cross-domain adaptation. Note that BOIL has achieved the best performance without the last skip connection in every experiment.

### 5.1 DISCONNECTION TRICK

Connecting the two learning schemes and ResNet's wiring structure, we propose a simple trick to eliminate the skip connection of the last residual block, which is referred to as a disconnection trick. In section 4.1, we confirmed that the model learned with BOIL applies the *representation layer reuse* at the low- and mid-levels of the body and *representation layer change* at the high-level of the body.

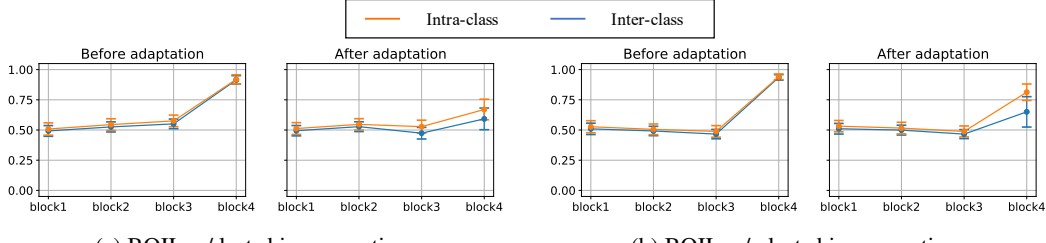

(a) BOIL w/ last skip connection.      (b) BOIL w/o last skip connection.

Figure 4: Cosine similarity of ResNet-12.

To investigate the effects of skip connections on a *representation change* learning scheme, we analyze the cosine similarity after every residual block in the same way as Figure 2. Figure 4a shows that ResNet with skip connections on all blocks rapidly changes not only the last block but also the other blocks. Because skip connections strengthen the gradient back-propagation, the scope of *representation layer change* extends to the front. Therefore, to achieve both the effective *representation layer reuse* and the *representation layer change* of BOIL, we suggest a way to weaken the gradient back-propagation from the loss function by removing the skip connection of the last block. As shown in Figure 4b, with this simple disconnection trick, ResNet can improve the effectiveness of BOIL, as well as the *representation layer reuse* at the front blocks of the body and the *representation layer change* at the last block, and significantly improves the performance, as described in Table 6.

We also report various analyses on ResNet-12 in the same way we analyzed 4conv network and *representation layer change* in the last block in Appendix M and Appendix N.

## 6 RELATED WORK

MAML (Finn et al., 2017) is one of the most well-known algorithms in gradient-based meta-learning, achieving a competitive performance on few-shot learning benchmark data sets (Vinyals et al., 2016; Ren et al., 2018; Bertinetto et al., 2018; Oreshkin et al., 2018). To tackle the task ambiguity caused by insufficient data in few-shot learning, numerous studies have sought to extend MAML in various ways. Some studies (Oreshkin et al., 2018; Sun et al., 2019; Vuorio et al., 2019) have proposed feature modulators that make task-specific adaptation more amenable by shifting and scaling the representations extracted from the network body. In response to the lack of data for task-specific updates, there have also been attempts to incorporate additional parameters in a small number, rather than all model parameters (Zintgraf et al., 2018; Rusu et al., 2018; Lee & Choi, 2018; Flennerhag et al., 2020). With a similar approach, some studies suggested a way to update only the heads in the inner loop, which has been further improved to update the head using linear separable objectives. (Raghu et al., 2020; Bertinetto et al., 2018; Lee et al., 2019). Grant et al. (2018); Finn et al. (2018); Yoon et al. (2018); Na et al. (2019) have taken a probabilistic approach using Bayesian modeling and variational inference. In addition, Chen et al. (2020) showed that by allowing the discovered task-specific modules (i.e., a (small) subset of a network) to adapt, better performance is achieved than when allowing the whole network to adapt. Notably, in few-shot image classification tasks, the authors showed that the last convolution layer (i.e., penultimate layer) is the most important. Such results were also observed in Arnold et al. (2019).

To tackle more realistic problems, few-shot learning has recently been expanding beyond the standard $n$-way $k$-shot classification. Triantafillou et al. (2019) constructed a more scalable and realistic data set, called a meta-data set, which contains several data sets collected from different sources. In additions, Na et al. (2019) addressed $n$-way any-shot classification by considering the imbalanced data distribution in real-world. Furthermore, some studies (Cai & Shen, 2020; Chen et al., 2019) have recently explored few-shot learning on cross-domain adaptation, which is one of the ultimate goals of meta-learning. In addition, Guo et al. (2019) suggested a new cross-domain benchmark data set for few-shot learning and showed that the current meta-learning algorithms (Finn et al., 2017; Vinyals et al., 2016; Snell et al., 2017; Sung et al., 2018; Lee et al., 2019) underachieve compared to simple fine-tuning on cross-domain adaptation. We demonstrated that task-specific update with *representation change* is efficient for a cross-domain adaptation.

## 7 CONCLUSION

In this study, we investigated the necessity of *representation change* for solving domain-agnostic tasks and proposed the BOIL algorithm, which is designed to enforce *representation change* by learning only the body of the model in the inner loop. We connected BOIL with preconditioning gradients and showed that the effectivenesses from a connection, such as an overfitting reduction and robustness to hyperparameters change, are still valid. Furthermore, we adapt BOIL to WarpGrad, demonstrating improved performance. This result decouples the benefits of *representation change* and preconditioning gradients. Next, we demonstrated that BOIL trains a model to follow the *representation layer reuse* scheme on the low- and mid-levels of the body but trains it to follow the *representation layer change* scheme on the high-level of the body using the cosine similarity and the CKA. We validated the BOIL algorithm on various data sets and a cross-domain adaptation using a standard 4conv network and ResNet-12. The experimental results showed a significant improvement over MAML/ANIL, particularly cross-domain adaptation, implying that *representation change* should be considered for adaptation to unseen tasks.

We hope that our study inspires *representation change* in gradient-based meta-learning approaches. Our approach is the first to study *representation change* and focuses on classification tasks. However, we believe that our approach is also efficient in other methods or fields because our algorithm has no restrictions. Furthermore, connecting *representation change* to memorization overfitting addressed in (Yin et al., 2019; Rajendran et al., 2020) will be an interesting topic.

### ACKNOWLEDGMENTS

This work was supported by Institute of Information & communications Technology Planning & Evaluation (IITP) grant funded by the Korea government(MSIT) (No.2019-0-00075, Artificial Intelligence Graduate School Program(KAIST)) and by Korea Electric Power Corporation (Grant number: R18XA05).

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

## A    IMPLEMENTATION DETAIL

### A.1    $n$-WAY $k$-SHOT SETTING

We experimented in the 5-way 1-shot and 5-way 5-shot, and the number of shots is marked in parentheses in the algorithm name column of all tables. During meta-training, models are inner loop updated only once, and the meta-batch size for the outer loop is set to 4. During meta-testing, the number of task-specific (inner loop) updates is the same as meta-training. All the reported results are based on the model with the best validation accuracy.

### A.2    MODEL IMPLEMENTATIONS

In our experiments, we employ the 4conv network and ResNet-12 for MAML/ANIL and BOIL algorithms. 4conv network has of 4 convolution modules, and each module consists of a $3 \times 3$ convolution layer with 64 filters, batch normalization (Ioffe & Szegedy, 2015), a ReLU non-linearity, a $2 \times 2$ max-pool. ResNet-12 (He et al., 2016) has the same structure with the feature extractor of TADAM (Oreshkin et al., 2018). It has four residual blocks, and each block consists of 3 modules of convolution, batch normalization, and leaky ReLU (Xu et al., 2015). At every end of each residual block, $2 \times 2$ max-pool is applied, and the number of convolution filters is doubled from 64 on each block. Each block also has a wiring structure known as skip connection, which is a link made up of additions between the block's input and output feature for strengthening feature propagation.

Our proposed algorithms can be implemented by just dividing learning rates into for the body and the head. Table 7 shows the learning rates of each network and algorithm. $\alpha_b$ and $\alpha_h$ are the learning rates of the body and the head of the model during inner loops, and $\beta_b$ and $\beta_h$ are the learning rates of the body and the head of the model during outer loops.

|          | 4conv network | | | ResNet-12 | | |
|----------|------|------|------|--------|--------|--------|
|          | MAML | ANIL | BOIL | MAML | ANIL | BOIL |
| $\alpha_b$ | 0.5 | 0.0 | 0.5 | 0.3 | 0.0 | 0.3 |
| $\alpha_h$ | 0.5 | 0.5 | 0.0 | 0.3 | 0.3 | 0.0 |
| $\beta_b$ | 0.001 | 0.001 | 0.001 | 0.0006 | 0.0006 | 0.0006 |
| $\beta_h$ | 0.001 | 0.001 | 0.001 | 0.0006 | 0.0006 | 0.0006 |

Table 7: Learning rates according to the algorithms.

### A.3    DATASET

We validate the BOIL and MAML/ANIL algorithms on several data sets, considering image size and fineness. Table 8 is the summarization of the used data sets.

Table 8: Summary of data sets.

| Data sets | miniImageNet | tieredImageNet | Cars | CUB |
|-----------|--------------|----------------|------|-----|
| Source | Russakovsky et al. (2015) | Russakovsky et al. (2015) | Krause et al. (2013) | Welinder et al. (2010) |
| Image size | 84×84 | 84×84 | 84×84 | 84×84 |
| Fineness | Coarse | Coarse | Fine | Fine |
| # meta-training classes | 64 | 351 | 98 | 100 |
| # meta-validation classes | 16 | 97 | 49 | 50 |
| # meta-testing classes | 20 | 160 | 49 | 50 |
| Split setting | Vinyals et al. (2016) | Ren et al. (2018) | Tseng et al. (2020) | Hilliard et al. (2018) |

| Data sets | FC100 | CIFAR-FS | VGG-Flower | Aircraft |
|-----------|-------|----------|------------|----------|
| Source | Krizhevsky et al. (2009) | Krizhevsky et al. (2009) | Nilsback & Zisserman (2008) | Maji et al. (2013) |
| Image size | 32×32 | 32×32 | 32×32 | 32×32 |
| Fineness | Coarse | Coarse | Fine | Fine |
| # meta-training classes | 60 | 64 | 71 | 70 |
| # meta-validation classes | 20 | 16 | 16 | 15 |
| # meta-testing classes | 20 | 20 | 15 | 15 |
| Split setting | Bertinetto et al. (2018) | Oreshkin et al. (2018) | Na et al. (2019) | Na et al. (2019) |

## B    VISUALIZATION USING UMAP

Through section 4.1, we show that conv4 in a 4conv network is the critical layer where *representation layer change* happens. We visualize these representations, the output of conv4, of samples from various data sets using UMAP (McInnes et al., 2018), which is an algorithm for general non-linear dimension reduction. *Samples with the same line color belong to the same class*. Many examples show the consistency with the intuition shown in Figure 1. When 1) similar instances with different classes are sampled together and 2) representations on the meta-train data set cannot capture representations on the meta-test data set, MAML/ANIL seems to be challenging to cluster samples on representation space since they are based on *representation reuse*.

## B.1 BENCHMARK DATA SETS

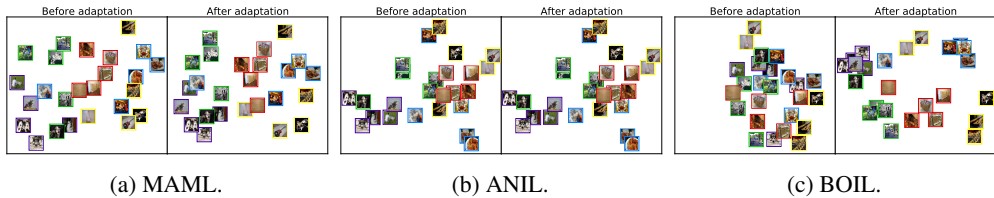

(a) MAML.  (b) ANIL.  (c) BOIL.

Figure 5: UMAP of samples from miniImageNet using the model meta-trained on miniImageNet.

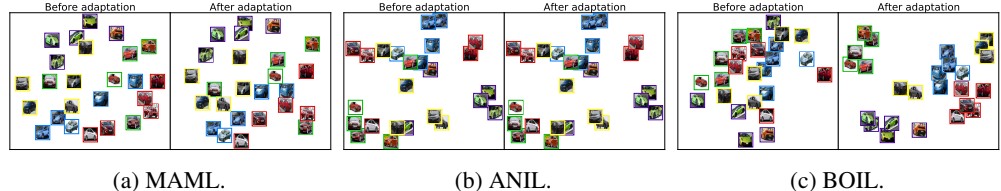

(a) MAML.  (b) ANIL.  (c) BOIL.

Figure 6: UMAP of samples from Cars using the model meta-trained on Cars.

## B.2 CROSS-DOMAIN ADAPTATION

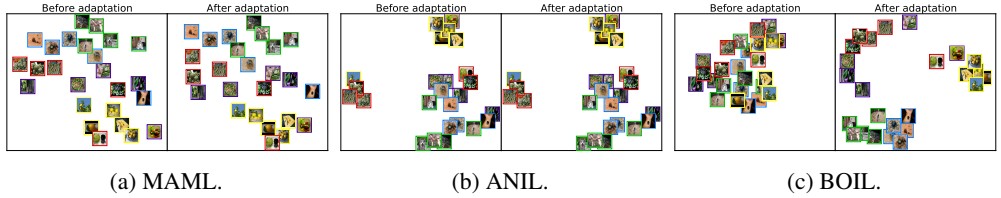

(a) MAML.  (b) ANIL.  (c) BOIL.

Figure 7: UMAP of samples from tieredImageNet using the model meta-trained on miniImageNet.

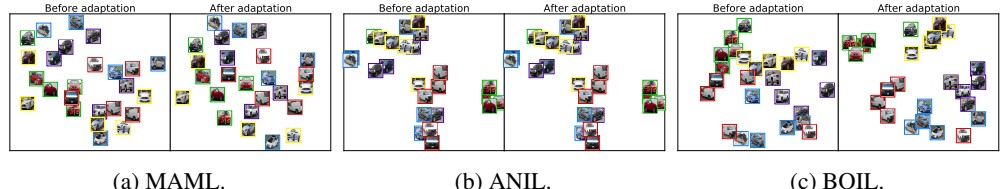

(a) MAML.  (b) ANIL.  (c) BOIL.

Figure 8: UMAP of samples from Cars using the model meta-trained on miniImageNet.

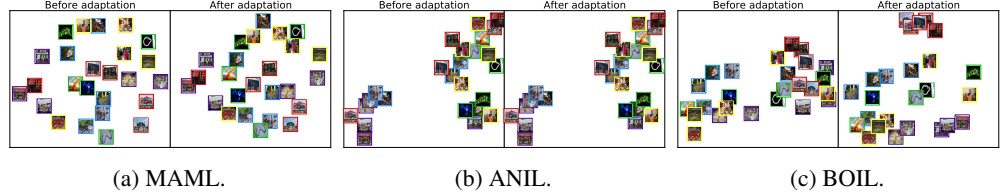

(a) MAML.  (b) ANIL.  (c) BOIL.

Figure 9: UMAP of samples from miniImageNet using the model meta-trained on Cars.

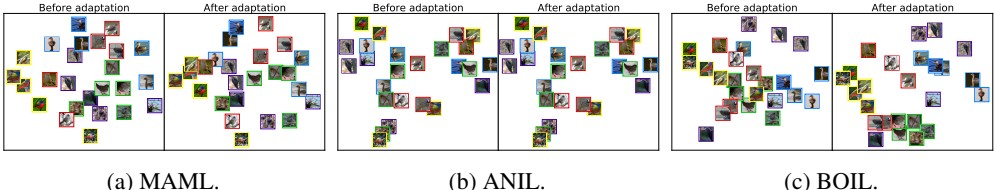

(a) MAML.  (b) ANIL.  (c) BOIL.

Figure 10: UMAP of samples from CUB using the model meta-trained on Cars.

## C Results on Other Data sets

We applied our algorithm to other data sets with image size of $32 \times 32$. Similar to the analyses on section 4, these data sets can be divided into two general data sets, **CIFAR-FS** (Bertinetto et al., 2018) and **FC100** (Oreshkin et al., 2018), and two specific data sets, **VGG-Flower** (Nilsback & Zisserman, 2008) and **Aircraft** (Maji et al., 2013). Table 9, Table 10, and Table 11 generally show the superiority of BOIL even if image size is extremely tiny.

Table 9: Test accuracy (%) of 4conv network on benchmark dataset.

| Domain | General (Coarse-grained) | | Specific (Fine-grained) | |
|---|---|---|---|---|
| Dataset | CIFAR-FS | FC100 | VGG-Flower | Aircraft |
| MAML(1) | $56.55 \pm 0.45$ | $35.99 \pm 0.48$ | $60.94 \pm 0.35$ | $52.27 \pm 0.23$ |
| ANIL(1) | $57.13 \pm 0.47$ | $36.37 \pm 0.33$ | $63.05 \pm 0.30$ | $\mathbf{54.54} \pm 0.16$ |
| BOIL(1) | $\mathbf{58.03} \pm 0.43$ | $\mathbf{38.93} \pm 0.45$ | $\mathbf{65.64} \pm 0.26$ | $53.37 \pm 0.29$ |
| MAML(5) | $70.10 \pm 0.29$ | $47.58 \pm 0.30$ | $75.13 \pm 0.43$ | $63.44 \pm 0.26$ |
| ANIL(5) | $69.87 \pm 0.39$ | $45.65 \pm 0.44$ | $72.07 \pm 0.48$ | $63.21 \pm 0.16$ |
| BOIL(5) | $\mathbf{73.61} \pm 0.32$ | $\mathbf{51.66} \pm 0.32$ | $\mathbf{79.81} \pm 0.42$ | $\mathbf{66.03} \pm 0.14$ |

Table 10: Test accuracy (%) of 4conv network on cross-domain adaptation.

| adaptation | general to general | | general to specific | | specific to general | | specific to specific | |
|---|---|---|---|---|---|---|---|---|
| meta-train | FC100 | CIFAR-FS | CIFAR-FS | CIFAR-FS | VGG-Flower | VGG-Flower | Aircraft | VGG-Flower |
| meta-test | CIFAR-FS | FC100 | VGG-Flower | Aircraft | CIFAR-FS | FC100 | VGG-Flower | Aircraft |
| MAML(1) | $62.58 \pm 0.35$ | $52.81 \pm 0.28$ | $49.69 \pm 0.24$ | $27.03 \pm 0.18$ | $34.38 \pm 0.19$ | $32.45 \pm 0.23$ | $37.05 \pm 0.19$ | $25.70 \pm 0.19$ |
| ANIL(1) | $\mathbf{63.05} \pm 0.39$ | $\mathbf{55.36} \pm 0.47$ | $50.61 \pm 0.29$ | $27.39 \pm 0.09$ | $35.90 \pm 0.20$ | $33.84 \pm 0.30$ | $31.59 \pm 0.22$ | $24.55 \pm 0.14$ |
| BOIL(1) | $60.71 \pm 0.43$ | $54.18 \pm 0.35$ | $\mathbf{56.77} \pm 0.35$ | $\mathbf{29.29} \pm 0.10$ | $\mathbf{39.15} \pm 0.20$ | $\mathbf{34.37} \pm 0.14$ | $\mathbf{49.85} \pm 0.26$ | $\mathbf{29.05} \pm 0.16$ |
| MAML(5) | $75.32 \pm 0.34$ | $63.00 \pm 0.18$ | $64.49 \pm 0.23$ | $33.85 \pm 0.25$ | $46.81 \pm 0.11$ | $42.06 \pm 0.43$ | $47.74 \pm 0.07$ | $30.65 \pm 0.19$ |
| ANIL(5) | $\mathbf{77.01} \pm 0.51$ | $63.89 \pm 0.16$ | $64.20 \pm 0.10$ | $33.24 \pm 0.21$ | $44.52 \pm 0.25$ | $40.51 \pm 0.26$ | $50.28 \pm 0.12$ | $28.74 \pm 0.23$ |
| BOIL(5) | $76.33 \pm 0.30$ | $\mathbf{68.55} \pm 0.20$ | $\mathbf{74.93} \pm 0.11$ | $\mathbf{39.96} \pm 0.11$ | $\mathbf{55.48} \pm 0.21$ | $\mathbf{47.17} \pm 0.38$ | $\mathbf{64.68} \pm 0.23$ | $\mathbf{39.81} \pm 0.25$ |

Table 11: 5-Way 5-Shot test accuracy (%) of ResNet-12. The lsc means the last skip connection.

| Meta-train | CIFAR-FS | | | VGG-Flower | | |
|---|---|---|---|---|---|---|
| Meta-test | CIFAR-FS | FC100 | VGG-Flower | VGG-Flower | CIFAR-FS | Aircraft |
| MAML w/ lsc | $75.30 \pm 0.19$ | $69.34 \pm 0.35$ | $65.82 \pm 0.30$ | $74.82 \pm 0.29$ | $42.91 \pm 0.20$ | $28.50 \pm 0.12$ |
| MAML w/o lsc | $71.72 \pm 0.19$ | $67.60 \pm 0.34$ | $59.20 \pm 0.26$ | $72.07 \pm 0.29$ | $39.27 \pm 0.23$ | $26.94 \pm 0.18$ |
| ANIL w/ lsc | $74.87 \pm 0.11$ | $75.34 \pm 0.45$ | $63.72 \pm 0.40$ | $77.02 \pm 0.29$ | $45.80 \pm 0.32$ | $27.24 \pm 0.13$ |
| ANIL w/o lsc | $71.39 \pm 0.28$ | $69.29 \pm 0.32$ | $52.70 \pm 0.24$ | $72.13 \pm 0.39$ | $38.99 \pm 0.22$ | $26.09 \pm 0.08$ |
| BOIL w/ lsc | $\mathbf{78.17} \pm 0.14$ | $\mathbf{77.22} \pm 0.45$ | $73.90 \pm 0.38$ | $82.00 \pm 0.17$ | $50.91 \pm 0.35$ | $35.54 \pm 0.25$ |
| BOIL w/o lsc | $77.38 \pm 0.10$ | $70.98 \pm 0.34$ | $\mathbf{73.96} \pm 0.27$ | $\mathbf{83.97} \pm 0.17$ | $\mathbf{55.82} \pm 0.44$ | $\mathbf{37.74} \pm 0.21$ |

## D Results under the original hyperparameters

We also evaluate our algorithm in the original setting (50 times smaller inner learning rate than ours) and confirm that BOIL is more robust to the change of hyperparameters than MAML. Such a characteristic is investigated in Lee & Choi (2018). Table 13 shows the test accuracy of BOIL and MAML/ANIL with the same hyperparameters optimized for MAML, and Figure 11 and Table 12 describe it according to the number of adaptation(s). It is observed that BOIL is the best or near-best, although the hyperparameters are not optimized for BOIL. Moreover, BOIL rapidly adapts and achieves considerable performance through just one adaptation.

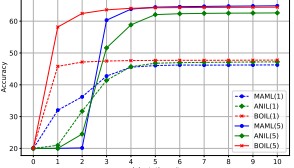

Figure 11: Accuracy on miniImageNet according to the number of adaptation(s).

Table 12: Test accuracy (%) according to the number of adaptation(s). Training and testing are on miniImageNet.

| Adaptation # | 1 | 2 | 3 | 4 | 5 | 6 | 7 | 8 | 9 | 10 |
|---|---|---|---|---|---|---|---|---|---|---|
| MAML(1) | $32.01 \pm 0.24$ | $36.21 \pm 0.17$ | $42.74 \pm 0.20$ | $45.54 \pm 0.18$ | $46.04 \pm 0.16$ | $46.21 \pm 0.19$ | $46.17 \pm 0.18$ | $46.20 \pm 0.18$ | $46.22 \pm 0.16$ | $46.25 \pm 0.18$ |
| ANIL(1) | $20.97 \pm 0.03$ | $31.68 \pm 0.25$ | $41.41 \pm 0.26$ | $45.69 \pm 0.20$ | $46.78 \pm 0.26$ | $46.95 \pm 0.30$ | $47.05 \pm 0.31$ | $47.10 \pm 0.30$ | $47.17 \pm 0.28$ | $47.20 \pm 0.27$ |
| BOIL(1) | $\mathbf{45.79} \pm 0.45$ | $\mathbf{47.15} \pm 0.30$ | $\mathbf{47.46} \pm 0.34$ | $\mathbf{47.61} \pm 0.34$ | $\mathbf{47.67} \pm 0.31$ | $\mathbf{47.70} \pm 0.32$ | $\mathbf{47.70} \pm 0.33$ | $\mathbf{47.71} \pm 0.34$ | $\mathbf{47.74} \pm 0.32$ | $\mathbf{47.76} \pm 0.31$ |
| MAML(5) | $20.02 \pm 0.00$ | $20.15 \pm 0.02$ | $60.31 \pm 0.34$ | $63.78 \pm 0.34$ | $\mathbf{64.41} \pm 0.35$ | $\mathbf{64.55} \pm 0.33$ | $\mathbf{64.64} \pm 0.31$ | $\mathbf{64.72} \pm 0.31$ | $\mathbf{64.77} \pm 0.30$ | $\mathbf{64.83} \pm 0.40$ |
| ANIL(5) | $20.00 \pm 0.00$ | $24.52 \pm 0.19$ | $51.59 \pm 0.17$ | $58.84 \pm 0.46$ | $62.06 \pm 0.37$ | $62.34 \pm 0.36$ | $62.45 \pm 0.38$ | $62.51 \pm 0.37$ | $62.55 \pm 0.38$ | $62.59 \pm 0.39$ |
| BOIL(5) | $\mathbf{58.15} \pm 0.23$ | $\mathbf{62.42} \pm 0.33$ | $\mathbf{63.56} \pm 0.26$ | $\mathbf{64.04} \pm 0.30$ | $64.21 \pm 0.28$ | $64.27 \pm 0.30$ | $64.32 \pm 0.30$ | $64.35 \pm 0.29$ | $64.38 \pm 0.28$ | $64.40 \pm 0.28$ |

Table 13: Test accuracy (%) under the same architecture, learning rate, and the number of inner updates with (Finn et al., 2017; Raghu et al., 2020).

| Meta-train | miniImageNet | | | Cars | | |
|---|---|---|---|---|---|---|
| Meta-test | miniImageNet | tieredImageNet | Cars | Cars | miniImageNet | CUB |
| MAML(1) | $46.25 \pm 0.18$ | $49.45 \pm 0.14$ | $34.78 \pm 0.36$ | $46.02 \pm 0.33$ | $28.87 \pm 0.11$ | $29.92 \pm 0.23$ |
| ANIL(1) | $47.20 \pm 0.27$ | $50.04 \pm 0.13$ | $32.87 \pm 0.39$ | $45.31 \pm 0.27$ | $29.12 \pm 0.11$ | $30.39 \pm 0.21$ |
| BOIL(1) | $\mathbf{47.76} \pm 0.31$ | $\mathbf{51.35} \pm 0.18$ | $\mathbf{34.89} \pm 0.23$ | $\mathbf{50.54} \pm 0.41$ | $\mathbf{32.40} \pm 0.19$ | $\mathbf{32.99} \pm 0.29$ |
| MAML(5) | $\mathbf{64.83} \pm 0.30$ | $\mathbf{67.06} \pm 0.25$ | $48.25 \pm 0.24$ | $\mathbf{69.27} \pm 0.27$ | $\mathbf{43.52} \pm 0.20$ | $45.12 \pm 0.20$ |
| ANIL(5) | $62.59 \pm 0.39$ | $65.55 \pm 0.16$ | $45.44 \pm 0.18$ | $62.67 \pm 0.25$ | $36.89 \pm 0.16$ | $40.38 \pm 0.19$ |
| BOIL(5) | $64.40 \pm 0.28$ | $65.81 \pm 0.26$ | $\mathbf{48.39} \pm 0.25$ | $68.56 \pm 0.34$ | $43.34 \pm 0.21$ | $\mathbf{46.32} \pm 0.11$ |

# E  OVERFITTING ISSUE

We employ networks with various sizes of filters, 32, 64, and 128. The best validation scores of each model are 64.01, 66.72, and 69.23, and these results mean that with BOIL, the more extensive network yields higher accuracy without overfitting.

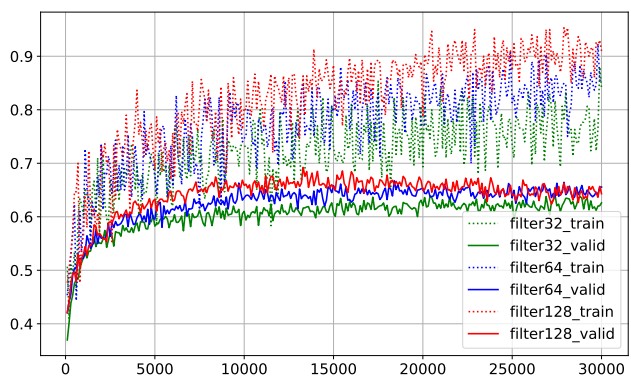

Figure 12: Training/Validation accuracy curve on miniImageNet according to filters in BOIL.

# F  WARPGRAD AND BOIL-WARPGRAD

## F.1  IMPLEMENTATION DETAIL

We follow the default setting of the public code except for meta train steps and the number of filters, following Flennerhag et al. (2020). We change meta train steps to 100 and the number of filters to 128. The task is 20way-5shot(in expectation) on Omniglot. Here, "in expectation" means that 100 samples are used for task-specific updates, but the number of samples per class is not the same. Furthermore, this task supports the superiority of BOIL in long-term adaptations.

## F.2  ARCHITECTURE

Here are the architectures of WarpGrad and BOIL-WarpGrad. The WarpGrad w/o last warp head model is the default one in the original code.

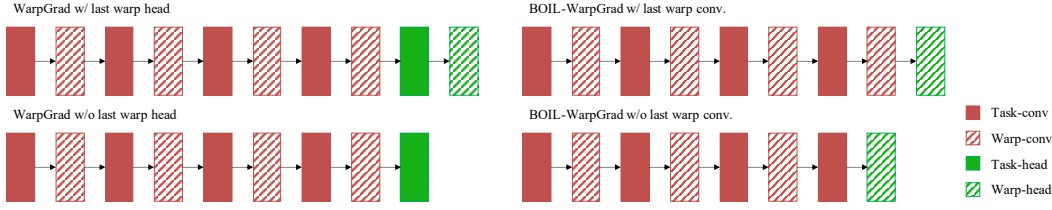

Figure 13: Architectures of WarpGrad and BOIL-WarpGrad.

# G  GRADIENT NORM

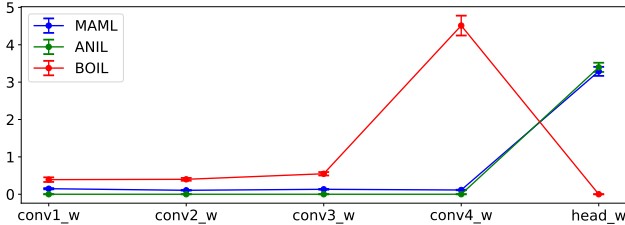

Figure 14: Gradient norm.

We calculated the norm of gradients caused by an inner loop according to the algorithm. Although the norm of gradients on the head of BOIL is not really zero, we marked 0 because the learning rate on the head is zero. The norm of gradients of biases is negligible (about $10^{-8}$), and thus it was omitted. MAML/ANIL has an extremely small norm or no norms on all convolutional layers. It implies that representations little or no change. On the other hand, BOIL has a large norm on the conv4 layer. It implies that representations change significantly. In addition, from the analysis of cosine similarity and CKA, we mentioned that BOIL enjoys *representation layer reuse* in a low- and mid-level of body. Nevertheless, Figure 14 shows that the amount of *representation layer change* in a low- and mid-level in BOIL is larger than that in MAML/ANIL.

# H  COSINE SIMILARITIES INCLUDING HEAD

Figure 15 is an extended version of Figure 2.

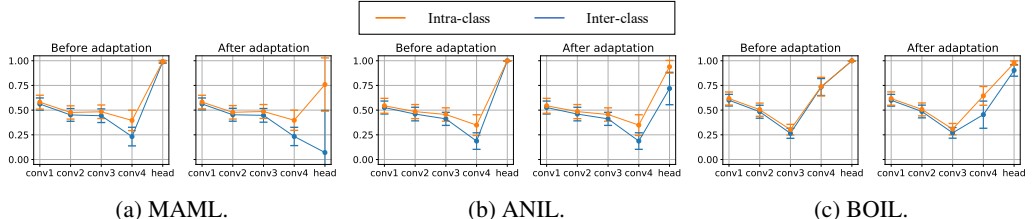

(a) MAML.         (b) ANIL.         (c) BOIL.

Figure 15: Cosine similarity of 4conv network including head.

# I  REPRESENTATION CHANGE IN BOIL ON CARS

This section describes *representation change* in BOIL on Cars. The structure of this section is the same as that of section 4.

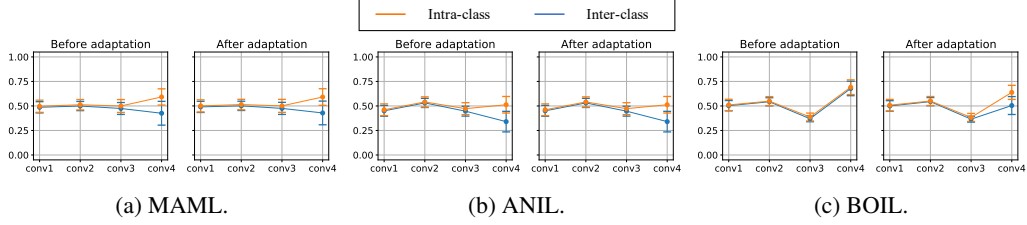

(a) MAML.         (b) ANIL.         (c) BOIL.

Figure 16: Cosine similarity of 4conv network on Cars.

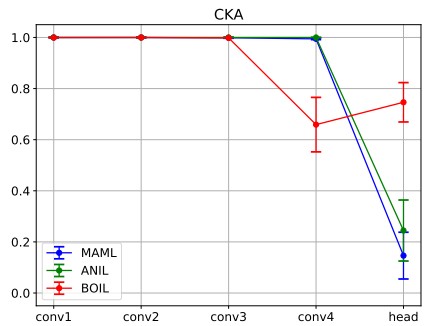

Figure 17: CKA of 4conv on Cars.

Table 14: Test accuracy (%) of 4conv network according to the head's existence before/after an adaptation.

| meta-train | Cars | | | | | | | |
|---|---|---|---|---|---|---|---|---|
| meta-test | Cars | | | | CUB | | | |
| head | w/ head | | w/o head (NIL-testing) | | w/ head | | w/o head (NIL-testing) | |
| adaptation | before | after | before | after | before | after | before | after |
| MAML(1) | 20.03 ± 0.25 | 45.27 ± 0.26 | 47.87 ± 0.18 | 47.23 ± 0.24 | 20.01 ± 0.08 | 29.64 ± 0.19 | **31.01 ± 0.26** | 31.15 ± 0.23 |
| ANIL(1) | 20.01 ± 0.18 | 46.81 ± 0.24 | **49.45 ± 0.18** | 49.45 ± 0.18 | 20.02 ± 0.08 | 28.32 ± 0.32 | 29.72 ± 0.27 | 29.72 ± 0.27 |
| BOIL(1) | 20.19 ± 0.19 | **56.82 ± 0.21** | 25.46 ± 0.29 | **52.36 ± 0.13** | 19.96 ± 0.12 | **34.79 ± 0.27** | 22.93 ± 0.17 | **34.51 ± 0.21** |
| MAML(5) | 20.11 ± 0.16 | 53.23 ± 0.26 | 59.67 ± 0.22 | 59.38 ± 0.23 | 20.00 ± 0.22 | 32.18 ± 0.13 | 36.12 ± 0.24 | 36.61 ± 0.19 |
| ANIL(5) | 20.09 ± 0.17 | 61.95 ± 0.38 | **67.03 ± 0.36** | 67.03 ± 0.36 | 19.99 ± 0.18 | 37.99 ± 0.15 | **43.27 ± 0.31** | 43.27 ± 0.31 |
| BOIL(5) | 20.04 ± 0.08 | **75.18 ± 0.21** | 36.65 ± 0.11 | **71.52 ± 0.27** | 20.02 ± 0.05 | **45.91 ± 0.28** | 29.04 ± 0.18 | **47.02 ± 0.25** |

## J OUTPUT LAYER OF MAML/ANIL AND PENULTIMATE LAYER OF BOIL

To investigate the role of the penultimate layer (i.e., the last convolutional layer) of BOIL, we evaluated MAML/ANIL and BOIL through NIL-testing on conv3 in the same way with NIL-testing on conv4 (Table 5), except for the position of representations. Table 15 shows that the input of the penultimate layer (i.e., features after conv3) cannot be simply classified (i.e., the desired performance cannot be achieved), and these representation capacities are similar for all MAML, ANIL, and BOIL. Therefore, It is thought that MAML/ANIL and BOIL acts similarly until conv3 and BOIL is not simply the shifted version of MAML/ANIL.

Table 15: Test accuracy (%) of NIL-testing on conv3 and conv4 of 4conv network before/after an adaptation.

| meta-train | miniImageNet | | | | | | | |
|---|---|---|---|---|---|---|---|---|
| meta-test | miniImageNet | | | | Cars | | | |
| head | NIL-testing on conv3 | | NIL-testing on conv4 | | NIL-testing on conv3 | | NIL-testing on conv4 | |
| adaptation | before | after | before | after | before | after | before | after |
| MAML(1) | 32.56 ± 0.14 | 32.73 ± 0.14 | 48.28 ± 0.20 | 47.87 ± 0.14 | 30.86 ± 0.09 | 31.03 ± 0.08 | 34.47 ± 0.19 | 34.36 ± 0.16 |
| ANIL(1) | **34.34 ± 0.10** | **34.34 ± 0.10** | **48.86 ± 0.12** | **48.86 ± 0.12** | **31.09 ± 0.08** | **31.09 ± 0.08** | **35.48 ± 0.24** | **35.48 ± 0.24** |
| BOIL(1) | 30.65 ± 0.07 | 31.09 ± 0.12 | 24.07 ± 0.19 | 46.73 ± 0.17 | 27.53 ± 0.16 | 27.80 ± 0.15 | 23.30 ± 0.15 | 34.07 ± 0.32 |
| MAML(5) | 51.03 ± 0.08 | 53.29 ± 0.10 | 64.61 ± 0.39 | 64.47 ± 0.39 | 44.54 ± 0.21 | 45.19 ± 0.21 | 47.66 ± 0.28 | 47.53 ± 0.28 |
| ANIL(5) | **55.55 ± 0.14** | **55.55 ± 0.14** | **66.11 ± 0.51** | **66.11 ± 0.51** | **45.81 ± 0.14** | **45.81 ± 0.14** | **49.62 ± 0.20** | 49.62 ± 0.20 |
| BOIL(5) | 49.42 ± 0.12 | 50.12 ± 0.12 | 32.03 ± 0.16 | 64.61 ± 0.27 | 43.93 ± 0.32 | 44.52 ± 0.18 | 30.33 ± 0.18 | **50.40 ± 0.30** |

Furthermore, the input of the output layer before adaptation (i.e., features after the penultimate layer) is enough to achieve the desired performance in MAML/ANIL in advance. From this result, we believe the head layer's role of MAML/ANIL is to draw a simple decision boundary for the high-level features represented by the output of the last convolutional layer. However, the penultimate layer of BOIL acts as a non-linear transformation so that the fixed output head layer can effectively conduct a classifier role.

# K   ABLATION STUDY ON THE LEARNING LAYER

In this section, we investigate whether training any representation layer during inner updates is better than training the output layer during inner updates by learning only one convolutional layer. Figure 18 shows the test accuracy according to a single learning layer in the body. For instance, conv1 plotted in red color indicates that the algorithm updates only the conv1 layer during inner updates. In contrast, conv1 plotted in blue color is the case that the algorithm updates the conv1 layer and the head layer during inner updates.

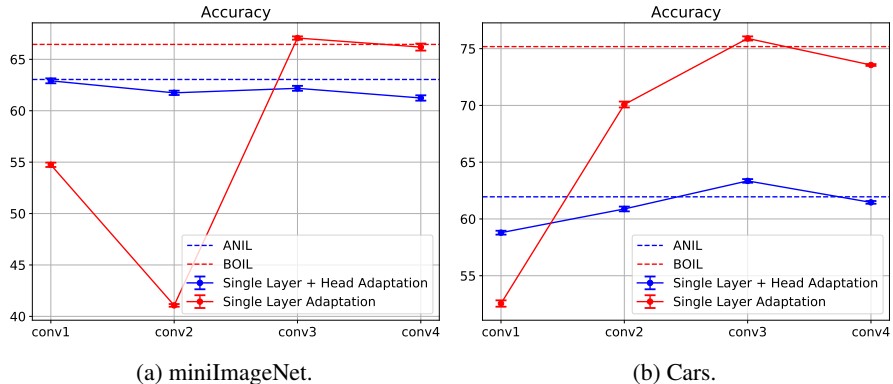

(a) miniImageNet.                                    (b) Cars.

Figure 18: Test accuracy according to the learning layer in the body.

On both miniImageNet and Cars, it is observed that training a higher-level representation layer (conv3, conv4) without updating the head (red line) performs better than training any single conv layer with the output layer (blue line). We also observe that learning only a lower-level representation layer (conv1, conv2) can significantly decrease accuracy. The results reassure that *representation layer change* in higher-level layers of the body boosts the performance discussed by the layer-wise analyses in section 4. However, training only a lower-level layer behaves badly since lower-level layers retain general representations (a related discussion is in Section 4.1, e.g., the third key observation).

Table 16: Test accuracy (%) of 4conv network according to the learning layer(s). Standard deviation is omitted.

| Learning layer | 1 layer | | | | 2 layers | | | | 3 layers | | | 4 layers | | all |
|---|---|---|---|---|---|---|---|---|---|---|---|---|---|---|
| conv1 | ✓ | | | | ✓ | | | | ✓ | | | ✓ | | ✓ |
| conv2 | | ✓ | | | ✓ | ✓ | | | ✓ | ✓ | | ✓ | ✓ | ✓ |
| conv3 | | | ✓ | | | ✓ | ✓ | | ✓ | ✓ | ✓ | ✓ | ✓ | ✓ |
| conv4 | | | | ✓ | | | ✓ | ✓ | | ✓ | ✓ | ✓ | ✓ | ✓ |
| head | | | | | ✓ | | | | ✓ | | | ✓ | | ✓ | ✓ |
| Algorithm | | | | ANIL | | | | | | | BOIL | | MAML |
| miniImageNet | 54.74 | 41.07 | **67.07** | 66.19 | 63.04 | 60.79 | **67.44** | 65.86 | 61.24 | 67.18 | **67.40** | 62.99 | **66.45** | 61.11 | 61.75 |
| tieredImageNet | 52.78 | 61.76 | 69.20 | **69.39** | 65.82 | 61.01 | 67.09 | **70.34** | 64.92 | 68.72 | **69.51** | 64.72 | **69.37** | 64.69 | 64.70 |
| Cars | 52.55 | 70.08 | **75.90** | 73.56 | 61.95 | 68.70 | **78.21** | 72.99 | 61.46 | 73.59 | **74.80** | 64.58 | **75.18** | 63.97 | 53.23 |
| CUB | 66.22 | 72.40 | **80.52** | 77.25 | 70.93 | 73.48 | **80.42** | 77.61 | 77.35 | **79.19** | 76.62 | 77.40 | **75.96** | 71.24 | 69.66 |
| CIFAR-FS | **71.58** | 70.47 | 71.01 | 70.88 | 69.87 | 72.15 | 71.85 | **74.43** | 71.43 | 72.93 | **75.56** | 72.07 | **73.61** | 71.73 | 70.10 |
| FC100 | 48.03 | 48.04 | **48.97** | 47.93 | 45.65 | 47.71 | 48.23 | **53.86** | 48.78 | 52.95 | **53.11** | 48.25 | **51.66** | 47.59 | 47.58 |
| VGG-Flower | **77.96** | 76.84 | 76.02 | 77.10 | 72.07 | 74.69 | 76.10 | **81.63** | 74.73 | **83.61** | 82.46 | 77.74 | **79.81** | 77.72 | 75.13 |
| Aircraft | 65.01 | 61.63 | 64.91 | **65.91** | 63.21 | 62.93 | 63.37 | **66.62** | 62.71 | 66.33 | **67.12** | 63.39 | **66.03** | 62.88 | 63.44 |

We expand this ablation study to training multiple consecutive layers with and without the head. The results are reported in Table 16 and Table 17. In Table 16, we consistently observe that learning with the head is far from the best accuracy. All the combinations having nice performances do not train the head in the inner loop update. We also find several settings skipping the lower-level layers in the inner loop that perform slightly better than BOIL. We believe each neural network architecture and data set pair has its own best layer combination. When it is allowed to search for the best combination using huge computing power, we can further improve BOIL. However, the most important design policy is that the inner loop update should freeze the head and encourage to learn higher-level representation features but to reuse lower-level representation features. BOIL follows the design rule by simply

Table 17: Test accuracy (%) of ResNet-12 without last skip connection according to the learning block(s). Standard deviation is omitted.

| Learning block | 1 block | | | | | 2 blocks | | | | 3 blocks | | | 4 blocks | | all |
|---|---|---|---|---|---|---|---|---|---|---|---|---|---|---|---|
| block1 | ✓ | | | | | ✓ | | | | ✓ | | | ✓ | | ✓ |
| block2 | | ✓ | | | | ✓ | ✓ | | | ✓ | ✓ | | ✓ | ✓ | ✓ |
| block3 | | | ✓ | | | | ✓ | ✓ | | ✓ | ✓ | ✓ | ✓ | ✓ | ✓ |
| block4 | | | | ✓ | | | | ✓ | ✓ | | ✓ | ✓ | ✓ | ✓ | ✓ |
| head | | | | | ✓ | | | | ✓ | | | ✓ | | ✓ | ✓ |
| Algorithm | | | | | ANIL | | | | | | | | BOIL | | MAML |
| miniImageNet | 19.94 | 64.20 | 69.95 | **70.52** | 67.20 | 63.08 | **69.19** | 67.75 | 66.19 | **70.12** | 69.76 | 69.08 | **71.30** | 66.44 | 67.87 |
| tieredImageNet | 20.10 | 47.64 | 68.57 | 70.41 | **72.22** | 55.22 | 69.69 | 70.11 | **72.38** | 67.64 | 69.61 | **71.67** | **73.44** | 71.02 | 71.25 |
| Cars | 19.98 | 55.86 | 74.71 | 74.45 | **75.32** | 65.30 | 70.41 | **74.06** | 69.51 | 68.16 | 58.43 | **69.78** | **83.99** | 71.75 | 73.63 |
| CUB | 20.02 | 74.84 | 79.26 | **81.58** | 74.66 | 75.07 | 80.12 | **82.09** | 74.24 | 80.26 | **82.75** | 74.94 | **83.22** | 75.66 | 76.23 |
| CIFAR-FS | 19.98 | 69.90 | 77.65 | **78.39** | 72.47 | 69.65 | 78.06 | **79.83** | 72.38 | 77.31 | **79.15** | 71.22 | **78.63** | 71.83 | 71.79 |
| FC100 | 20.15 | 48.32 | **50.86** | 49.60 | 45.61 | 47.44 | 51.75 | **51.82** | 46.93 | **50.72** | 50.55 | 45.21 | **49.87** | 46.29 | 44.90 |
| VGG-Flower | 19.98 | 80.32 | **84.68** | 82.22 | 73.77 | 79.80 | **85.13** | 80.75 | 72.14 | **83.53** | 82.69 | 71.93 | **82.17** | 72.00 | 72.43 |
| Aircraft | 20.06 | 71.48 | 76.97 | 77.22 | **78.62** | 72.17 | 78.47 | 77.61 | **79.51** | 76.75 | 76.89 | **78.16** | **78.85** | 78.79 | 77.15 |

freezing the head in the inner loop that is already almost the best approach in most cases. In Table 17, there are the cases where learning a classifier leads to performance improvement. We thought that this is because the ablation study on ResNet-12 is done at the block-level. More precisely, one block includes many layers and this issue is discussed in Appendix N.

## L  ADDITIONAL CONSIDERATIONS OF THE HEAD OF BOIL

We additionally discuss what the ideal meta-initialization is. Because the few-shot classification tasks are constructed with sampled classes each time, every task consists of different classes. Since the class indices are randomly assigned at the beginning of each task learning, the meta-initialized parameters cannot contain any prior information on the class indices. For instance, it is not allowed that the meta-initialized parameters encode class similarities between class $i$ and class $j$. Any biased initial guess could hinder the task learning. The meta-initialized parameters should be in-between (local) optimal points of tasks as depicted in Figure 19 so that the network can adapt to each task with few task-specific updates.[6]

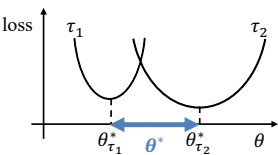

Figure 19: Ideal meta-initialization.

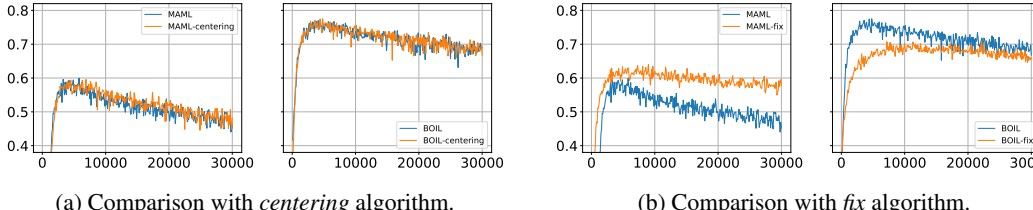

(a) Comparison with *centering* algorithm.  (b) Comparison with *fix* algorithm.

Figure 20: Valid accuracy curves of (a) centering algorithm and (b) fix algorithm on Cars.

When the head parameters $\theta_h = [\theta_{h,1}, ..., \theta_{h,n}]^\top \in \mathbb{R}^{n \times d}$ have orthonormal rows (i.e., $\|\theta_{h,i}\|_2 = 1$ for all $i$ and $\theta_{h,i}^\top \theta_{h,j} = 0$ for all $i \neq j$), the meta-initialized model can have the unbiased classifier. Here, $a^\top$ denotes the transpose of $a$ and $\| \cdot \|_2$ denotes the Euclidean norm. With the orthonormal rows, therefore, each logit value $\theta_{h,j}^\top f_{\theta_b}(x)$ can be controlled independently of other logit values. Recall that the softmax probability $p_j$ for class $j$ of sample $x$ is computed as follows:

$$p_j(x) = \frac{e^{\theta_{h,j}^\top f_{\theta_b}(x)}}{\sum_{i=1}^{n} e^{\theta_{h,i}^\top f_{\theta_b}(x)}} = \frac{1}{\sum_{i=1}^{n} e^{(\theta_{h,i} - \theta_{h,j})^\top f_{\theta_b}(x)}}. \tag{4}$$

---

[6]The similar consideration is discussed in (Drumond et al., 2020).

In Equation 4, indeed, the softmax probability only depends on the differences of the rows of the head parameters $\theta_{h,i} - \theta_{h,j}$. Adding a vector to all the rows (i.e., $\theta_{h,i} \leftarrow \theta_{h,i} + c$ for all $i$) does not change the softmax vector. So, we can expect the same nice meta-initialized model, when a parallel shift of the rows of the head parameters can make orthonormal rows. To support this experimentally, we design the *centering* algorithm that operates a parallel shift of $\theta_h$ by subtracting the average of the row vectors of $\theta_h$ after every outer update on both MAML and BOIL, i.e., $[\theta_{h,1} - \bar{\theta}_h, ..., \theta_{h,n} - \bar{\theta}_h]^\top$ where $\bar{\theta}_h = \frac{1}{n}\sum_{i=1}^{n}\theta_{h,i}$. Figure 20a shows that this parallel shift operations does not affect the performance of two algorithms on Cars.

Next, we investigate the cosine similarity between $\theta_{h,i}^\top - \theta_{h,k}^\top$ and $\theta_{h,j}^\top - \theta_{h,k}^\top$ for all different $i$, $j$, and fixed $k$. From the training procedures of MAML and BOIL, it is observed that the average of cosine similarities between the two gaps keeps near 0.5 during meta-training (Figure 21). Note that 0.5 is the cosine similarity between $\theta_{h,i}^\top - \theta_{h,k}^\top$ and $\theta_{h,j}^\top - \theta_{h,k}^\top$ when $\theta_{h,i}^\top$, $\theta_{h,j}^\top$, and $\theta_{h,k}^\top$ are orthonormal. From the results, we evidence that the orthonormality of $\theta_h$ is important for the meta-initialization and meta learning algorithms naturally keep the orthonormality.

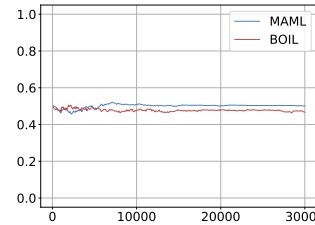

Figure 21: Average of cosine similarities between gaps.

From the above observation, we design the *fix* algorithm that fixes $\theta_h$ to be orthonormal for the meta-initialized model. Namely, MAML-fix updates $\theta_h$ in inner loops only, and BOIL-fix does not update $\theta_h$. The *fix* algorithm can be easily implemented by initializing $\theta_h$ to be orthonormal through the Gram-Schmidt method from a random matrix and setting the learning rate for the head of the model during the outer loop to zero.

Figure 20b depicts the valid accuracy curves of the *fix* algorithm on Cars. The experiments substantiate that orthonormal rows of $\theta_h$ are important and that BOIL improves the performance. (1) Comparing MAML to MAML-fix (the left panel of Figure 20b), MAML-fix outperforms MAML. It means that the outer loop calculated through the task-specific head following MAML is detrimental because the outer loop adds unnecessary task-specific information to the model. (2) Comparing vanilla models to fix models (both panels of Figure 20b), a fixed meta-initialized head with orthonormality is less over-fitted. (3) Comparing BOIL to BOIL-fix (the right panel of Figure 20b), although BOIL-fix can achieve almost the same performance with BOIL with sufficient iterations, BOIL converges faster to a better local optimum. This is because $\theta_h$ is trained so that the inner loop can easily adapt $f_{\theta_b}(x)$ to each class.

## M    REPRESENTATION CHANGE IN RESNET-12

Figure 22 shows the CKA of ResNet according to the algorithm. Like the 4conv network, MAML/ANIL algorithms change the values only in the logit space, i.e., the space after head, regardless of the last skip connection. However, the BOIL algorithm changes the values in the representation spaces. By disconnecting the last skip connection, *representation layer change* is concentrated on the high-level representation space, i.e., the CKA of BOIL w/o LSC is smaller than that of BOIL w/ LSC after block4. Table 18 shows empirical results of ResNet-12.

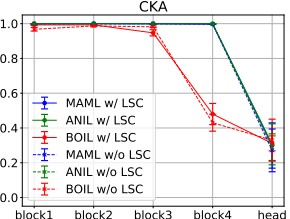

Figure 22: CKA of ResNet-12 on miniImageNet.

Table 18: 5-Way 5-Shot test accuracy (%) of ResNet-12 meta-trained on miniImageNet according to the head's existence before/after an adaptation.

| meta-train | miniImageNet | | | | | | | |
|---|---|---|---|---|---|---|---|---|
| meta-test | miniImageNet | | | | Cars | | | |
| head | w/ head | | w/o head (NIL-testing) | | w/ head | | w/o head (NIL-testing) | |
| adaptation | before | after | before | after | before | after | before | after |
| MAML w/ LSC | $20.02 \pm 0.21$ | $68.51 \pm 0.39$ | $\mathbf{70.44 \pm 0.30}$ | $70.37 \pm 0.32$ | $20.06 \pm 0.07$ | $43.46 \pm 0.15$ | $46.08 \pm 0.22$ | $46.05 \pm 0.19$ |
| MAML w/o LSC | $20.04 \pm 0.36$ | $67.87 \pm 0.22$ | $69.35 \pm 0.15$ | $69.28 \pm 0.14$ | $19.98 \pm 0.16$ | $41.40 \pm 0.11$ | $43.55 \pm 0.17$ | $43.56 \pm 0.16$ |
| ANIL w/ LSC | $19.85 \pm 0.19$ | $68.54 \pm 0.34$ | $70.31 \pm 0.34$ | $70.31 \pm 0.34$ | $19.99 \pm 0.15$ | $45.13 \pm 0.15$ | $\mathbf{47.16 \pm 0.20}$ | $47.16 \pm 0.20$ |
| ANIL w/o LSC | $19.97 \pm 0.21$ | $67.20 \pm 0.13$ | $68.47 \pm 0.21$ | $68.47 \pm 0.21$ | $20.03 \pm 0.23$ | $43.46 \pm 0.18$ | $45.16 \pm 0.11$ | $45.16 \pm 0.11$ |
| BOIL w/ LSC | $20.01 \pm 0.18$ | $70.50 \pm 0.28$ | $44.65 \pm 0.49$ | $70.34 \pm 0.31$ | $20.02 \pm 0.08$ | $49.69 \pm 0.17$ | $38.73 \pm 0.16$ | $49.54 \pm 0.23$ |
| BOIL w/o LSC | $20.00 \pm 0.00$ | $\mathbf{71.30 \pm 0.28}$ | $40.04 \pm 0.33$ | $\mathbf{71.18 \pm 0.29}$ | $20.00 \pm 0.00$ | $\mathbf{49.71 \pm 0.28}$ | $32.53 \pm 0.29$ | $\mathbf{51.41 \pm 0.32}$ |

Furthermore, we identified *representation change* of ResNet-12 meta-trained on Cars in BOIL.

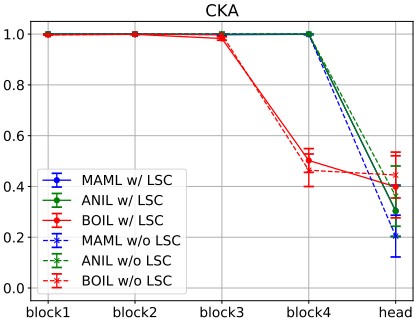

| | |
| --- | --- |
| (a) MAML w/ LSC. | |

(a) MAML w/ LSC.   (b) ANIL w/ LSC.   (c) BOIL w/ LSC.

(d) MAML w/o LSC.   (e) ANIL w/o LSC.   (f) BOIL w/o LSC.

Figure 23: Cosine similarity of ResNet-12 on Cars.

Figure 24: CKA of ResNet-12 on Cars.

Table 19: 5-Way 5-Shot test accuracy (%) of ResNet-12 meta-trained on Cars according to the head's existence before/after an adaptation.

| meta-train | Cars | | | | | | | |
| --- | --- | --- | --- | --- | --- | --- | --- | --- |
| meta-test | Cars | | | | CUB | | | |
| head | w/ head | | w/o head (NIL-testing) | | w/ head | | w/o head (NIL-testing) | |
| adaptation | before | after | before | after | before | after | before | after |
| MAML w/ LSC | $20.11 \pm 0.22$ | $75.49 \pm 0.20$ | $77.01 \pm 0.14$ | $76.76 \pm 0.17$ | $20.13 \pm 0.10$ | $35.87 \pm 0.19$ | $36.57 \pm 0.20$ | $36.62 \pm 0.17$ |
| MAML w/o LSC | $20.04 \pm 0.08$ | $73.63 \pm 0.26$ | $75.81 \pm 0.20$ | $75.61 \pm 0.25$ | $20.03 \pm 0.14$ | $34.77 \pm 0.26$ | $36.26 \pm 0.21$ | $36.04 \pm 0.27$ |
| ANIL w/ LSC | $20.06 \pm 0.35$ | $79.45 \pm 0.23$ | $\mathbf{80.88} \pm 0.19$ | $80.88 \pm 0.19$ | $20.12 \pm 0.14$ | $35.09 \pm 0.19$ | $35.80 \pm 0.16$ | $35.80 \pm 0.16$ |
| ANIL w/o LSC | $20.20 \pm 0.33$ | $75.32 \pm 0.15$ | $76.90 \pm 0.16$ | $76.90 \pm 0.16$ | $20.01 \pm 0.18$ | $36.06 \pm 0.14$ | $\mathbf{37.34} \pm 0.13$ | $37.34 \pm 0.13$ |
| BOIL w/ LSC | $20.00 \pm 0.00$ | $80.98 \pm 0.14$ | $40.54 \pm 0.11$ | $81.11 \pm 0.24$ | $20.00 \pm 0.00$ | $43.84 \pm 0.21$ | $37.27 \pm 0.12$ | $\mathbf{45.44} \pm 0.23$ |
| BOIL w/o LSC | $20.00 \pm 0.00$ | $\mathbf{83.99} \pm 0.20$ | $50.42 \pm 0.23$ | $\mathbf{83.60} \pm 0.18$ | $20.00 \pm 0.00$ | $\mathbf{44.23} \pm 0.18$ | $31.69 \pm 0.14$ | $44.19 \pm 0.17$ |

# N    REPRESENTATION LAYER CHANGE IN THE LAST BLOCK OF RESNET-12

In this section, we explore the *representation layer reuse* and *representation layer change* in the last block of a deeper architecture network. Figure 25 and Figure 26 show the cosine similarities between representations after all layers in the last block (i.e., block4) on miniImageNet and Cars. In the case of MAML/ANIL, there is little or no *representation layer change* in all layers of the last block. In contrast, in the case of BOIL, the gap between intra-class similarity and inter-class similarity is enlarged through adaptation in some or all layers of the last block, which indicates that *representation layer reuse* in low-level layers of the last block and *representation layer change* in high-level layers of the block are mixed even in the last block.

Furthermore, it is observed that representation at high-level layers in the last block changes more when the last skip connection does not exist (e.g., Figure 25f and Figure 26f) than when the last

skip connection exists (e.g., Figure 25e and Figure 26e). This result confirms that the disconnection trick strengthens *representation layer change* at high-level layers by not directly propagating general representations from prior blocks.

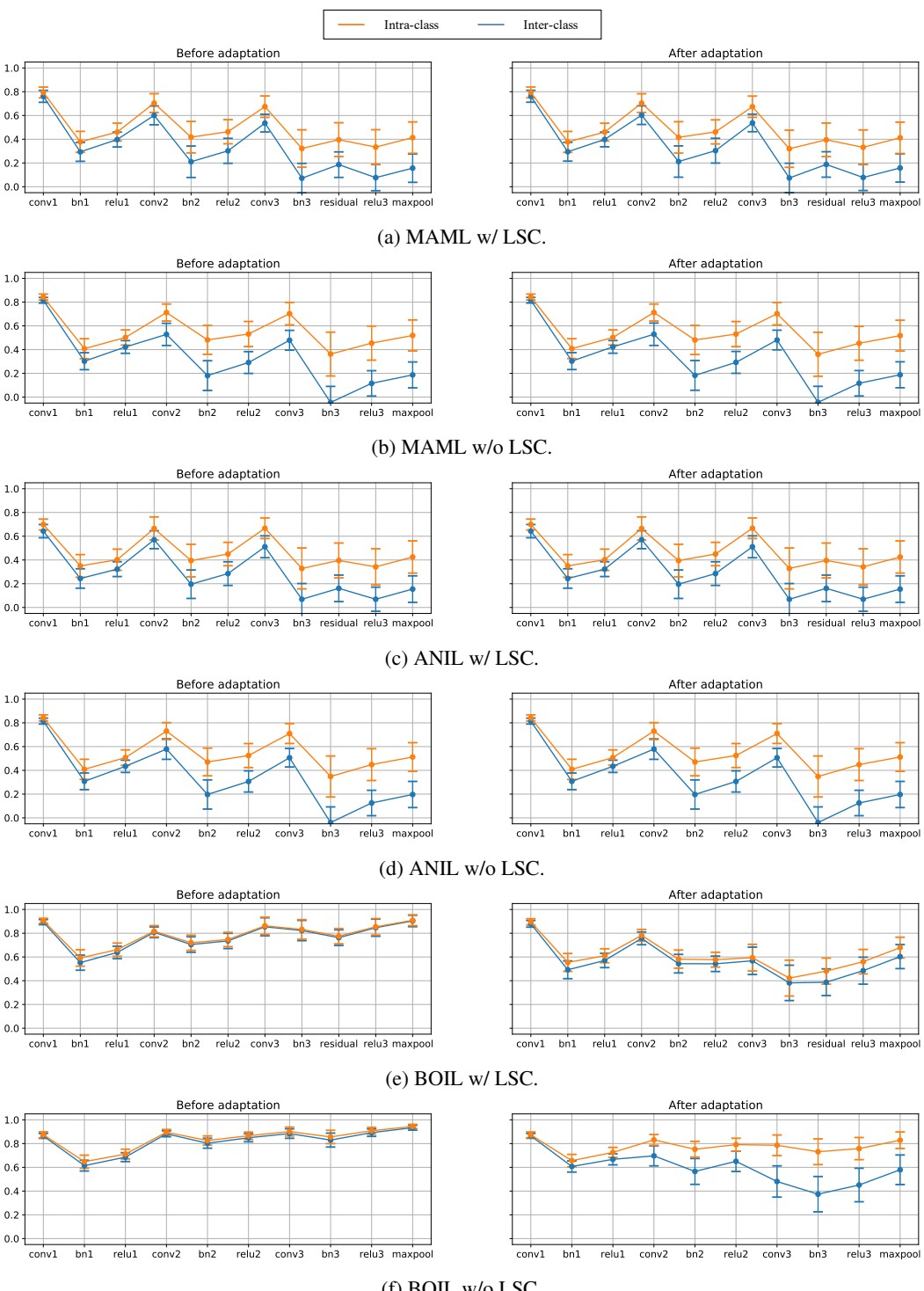

Figure 25: Cosine similarity in block4 (the last block) of ResNet-12 on miniImagenet.

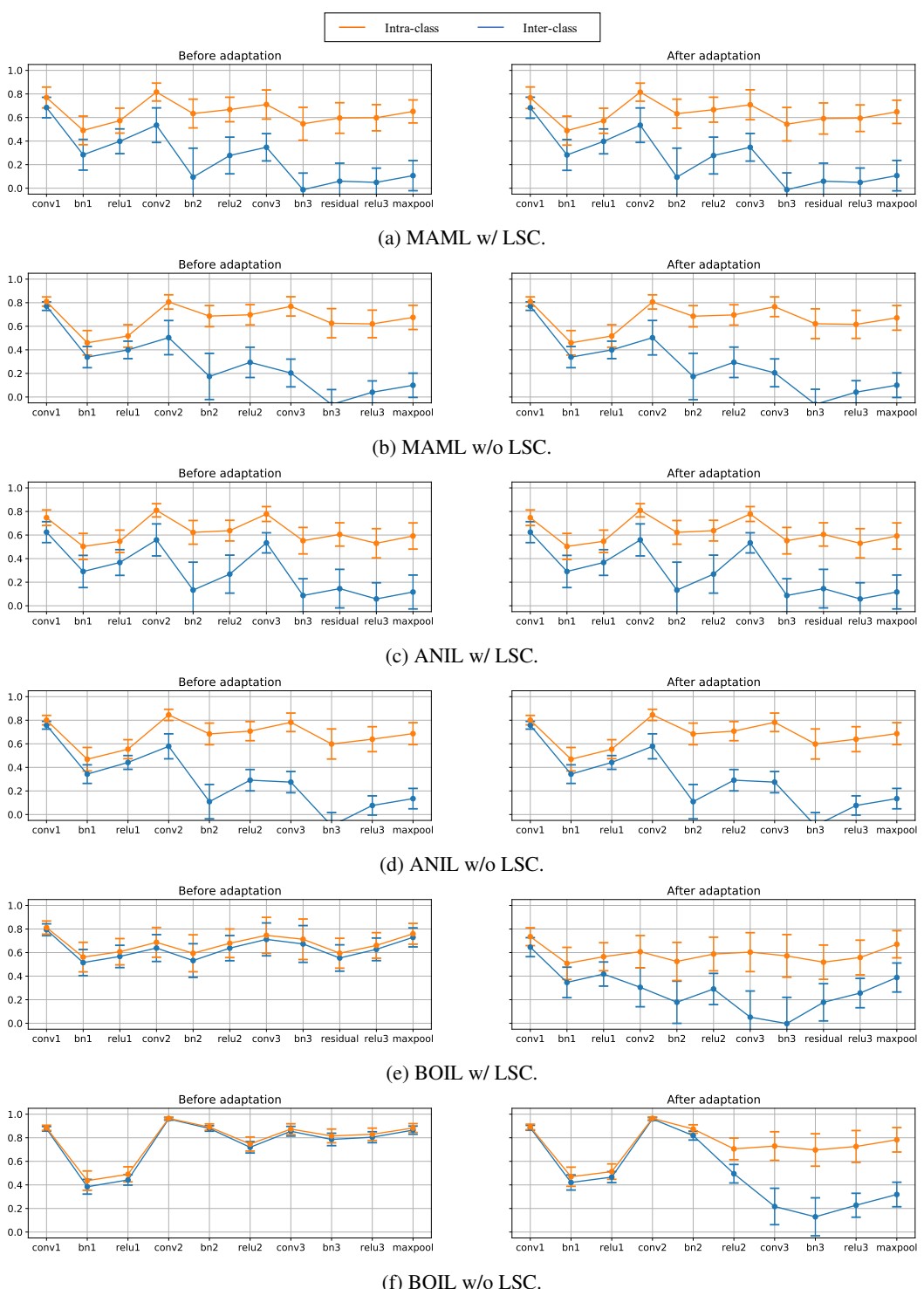

Figure 26: Cosine similarity in block4 (the last block) of ResNet-12 on Cars.

