# OpenReview forum: "BOIL: Towards Representation Change for Few-shot Learning"
_ICLR.cc/2021/Conference — ICLR 2021 Poster_

### Official Review · AnonReviewer4 · 2020-10-27

**Rating:** 7
**Confidence:** 5

**Review:**

#### Summary:
This paper proposes a variant of MAML, called body only in the inner loop (BOIL), in which the output layer is not trained (learning rate is set to 0, always) but gradients are still backpropagated to the remaining layers, which are trained as in MAML. In essence, this is the inverse of the ANIL method of Raghu et al., which showed that MAML mainly adapts only the output layer for few-shot image classification. This paper shows that never training the output layer of the network (simply leaving it at its randomly-initialized values) can provide much better performance across a number of few-shot image classification tasks, including those requiring domain transfer.

Pros:
- Consistently strong performance across a number of tasks
- Simple method
- Many experiments showing similar benefits
- Side studies and ablations to try to better understand the reason for the improved performance provided by BOIL

Cons:
- I am not convinced about the explanation of representation change versus representation reuse. Clearly, a lot of representation reuse is still occurring; i.e., all but one layer. The only difference is that instead of the final layer adapting and the remaining layers staying fixed, as in MAML, the final layer is fixed and the penultimate layer (now the final adapting layer) adapts. The preceding layers remain essentially fixed however (i.e., are reused).
- The experiments are somewhat unfair. Instead of performing a hyperparameter search for each method to choose the best settings for each, a single setting is used for all methods. While there is a lot of overlap between the methods, it’s clear from Appendix E that MAML can perform better using those hyperparameters in some domains than is reported in the main text. While it’s likely that BOIL will still do well (as it still does better than the improved MAML scores), it would be more fair to report the higher score for MAML. In particular, it’s likely that the much lower learning rate (as used in Appendix E) prevents the severe overfitting commonly observed in MAML.

#### Decision:
Overall, I am on the fence about this paper. The results are strong and will be interesting to the community; however, the explanation does not seem valid to me and the evaluation could be more fair. I thus recommend a weak accept as readers will be interested in the results and can draw their own conclusions about the reasons for them.

#### Questions:

1. Why does adapting only the penultimate layer count as representation change but adapting only the output layer is representation reuse? This distinction seems arbitrary. Clearly, there is a lot of benefit to adapting the penultimate layer as shown by the experiments but this seems like an identical amount of representation reuse as MAML with a network that is missing a layer and instead has a random transform appended to the end. How is this different from that?

2. What does Figure 2 look like if the cosine similarity of the output layer is plotted in each subplot along with that of the conv layers? Does BOIL then become a shifted version of MAML, as Figure 3 indicates it might?

#### Comments:
- A recent work [1] also examines fixing some layers, including the output layer, to obtain better performance than allowing all layers to adapt. It should be cited and discussed in the related work as well.
- Please define $f$ in equation (1).
- The final sentence of Section 3 is unintelligible to me. Please clarify it.
- In Figure 2, the subfigures are far too small to read. Further, the subtitles say “adatation” instead of “adaptation”.
- In general, the writing could use more editing.
- The whitespace around the titles is overly compressed. Please move some text to the Appendix instead of cramming as much as possible into the main body.

[1] Modular Meta-Learning with Shrinkage. Chen, Friesen, Behbani, Doucet, Budden, Hoffman, and de Freitas.


***********

After the discussion period, I have increased my score to 7 as the authors have provided a strong set of ablations and changes to address my concerns and those of the other reviewers.

---

> ### Author Response · Authors · 2020-11-15
> **Replies for your valuable review**
>
> Thank you for your careful review of our paper and for the insightful and constructive comments. We have addressed your comments and updated our draft accordingly. Please find our detailed answers below.
>
> (Cons)
> 1.	We re-expressed "feature reuse" and "rapid learning" from Raghu et al. (2020) to "representation reuse" and "representation change" for a direct delivery of the meaning (Section 1, annotation 1). In their study, these two terms are defined at the algorithm-level, e.g., "MAML is based on representation reuse." \
> We agree that the adaptations of BOIL should be explained in a layer-wise manner. We now specify the level of the terminologies from the algorithm-level to the layer-level in the extractor. We added an additional explanation about this issue (Section 4.1, annotation 5). With the new definitions, MAML is based on representation reuse on all layers in the extractor. By contrast, BOIL utilizes representation reuse in low- and mid-level layers and representation change in the highest-level layer in the extractor.
>
> 2.	We thought that describing the results under the same setting in a single table makes it less confusing. We have added annotations related to this issue (Section 2.2, annotation 3).
>
> (Questions)
> 1.	Representation reuse or representation change is defined with spaces related to representations, not related to logits. Adapting only the penultimate layer changes only the “representation,” and adapting only the output layer changes the “logit.” In addition, we have added a new graph on the gradient norm to Appendix I that can compare the amount of representation reuse in BOIL and MAML/ANIL. We observed that the gradient norms on conv1-conv4 in BOIL are much greater than those in MAML/ANIL. Based on the results, we believe that BOIL changes the representations more than MAML/ANIL at every layer except the output head layer. \
> MAML with a random transform differs from BOIL. This is because MAML updates the output layer during inner updates and BOIL learns the linear transform only through the outer updates. We investigated MAML with a random transform appended to the end, which is coined as the MAML-fix algorithm, where all parameters of the model are updated during the inner loops and the output layer is reset to the random initial point after the outer loops (please refer to Appendix J). It is observed that BOIL again overwhelms MAML-fix.
>
> 2.	We have added more cosine similarity graphs that include the output layer as well (Appendix G). As you noted, the change in difference between inter-class and intra-class on the head in MAML/ANIL moves to the change in difference between inter-class and intra-class on conv4 in BOIL. However, we believe that the learning pattern of BOIL is not a simple layer shift (Appendix L). We would also like to point the reviewer to our answer #1 for AnonReviewer #1.
>
> (Comments)
> We checked all of your comments and reflected them in our revised version of the manuscript.
> 1.	Thank you for informing us about a closely related study. They showed that the penultimate layer is the most task-specific layer in few-shot classification tasks (Appendix E.4.3. in the recommended paper). We added a discussion of a related paper (third bullet point in Section 4.1 and Section 6).
> 2.	We added the definition of f_\theta (Section 2.1).
> 3.	In Flennerhag et al. (2020), the authors inserted a warping layer (i.e., a fixed layer during inner updates) after all convolution layers. However, we observed that the penultimate layer changed significantly in BOIL (Figures 2 and 3), and thus we removed the last warping convolutional layer (BOIL-WarpGrad without the last warp conv in Table 3). From the results, we identified that, to support BOIL, the last convolution layer must not be fixed during the inner loops. We also clarified the sentence (Section 3.2.2).
> 4.	We resized the figures and modified the typing errors.
> 5.	We had the paper proofread by a professional editing service.
> 6.	We resolved this issue by using an additional page (nine pages are allowed during the rebuttal phase).

---

> > ### Comment · AnonReviewer4 · 2020-11-23
> > **Clarification and a further ablation to better understand the method**
> >
> > Thank you for your response and your updates to the paper. Let me try to clarify my position based on your response. From the results in the paper, it is clear that fixing the output layer and training the body of the network outperforms both traditional MAML and ANIL. This is a very interesting finding. However, this paper does not really tell me _why_ (in an empirically validated and meaningful way) we observe this behavior. Yes, the paper defines "representation learning" vs "representation reuse" but these terms do not actually inform us in a meaningful way as to why this performance boost occurs. This is because as I and the other reviewers have pointed out, most "representation" layers are still "reused". So the paper is really about how forcing reuse of the output layer and the resulting adaptation of the penultimate layer can greatly improve performance. Yes, this penultimate layer is a representation layer, however, most representation layers actually are reused so the choice of naming is more confusing than helpful.
> >
> > If it is true that training any representation layer is better than training the output layer, then an interesting ablation would be to perform ANIL/BOIL but only train one representation layer, and then do this for each of the representation layers. i.e., run 4 separate experiments where in experiment $i \in \{1, 2, 3, 4\}$ you train only layer $\textrm{conv}_i$ and all other layers have learning rate $0$. This would be an interesting ablation that could tell us whether _adapting the representation layers_ is what is important vs. whether _not adapting the output layer_ is what is important. The paper postulates the former but the results only speak to the latter so far.
> >
> > Another ablation that might help establish _why_ training the representation layers is important (as I'm still not sure from this paper but would really like to know) and would be very informative is to run BOIL but not set the learning rate of the output layer to $0$, instead sweep over values of it (from $0$ to the overall learning rate). This would paint an interesting picture. If training the head is bad, then $0$ should perform best and performance decrease as the learning rate of the head increases. However, if instead the relatively poor performance of MAML here is due to overfitting caused by adapting the output layer, then some small but non-zero learning rate should perform best.
> >
> > I understand you may not have time to run all of these ablations but I encourage you to run them and include them if possible before the discussion period. Failing that, to include them in the next iteration of the paper.

---

> > > ### Comment · AnonReviewer1 · 2020-11-24
> > > **Agree with this summary by Reviewer4**
> > >
> > > This is a great summary of my main concern also, and I refer to it in my response.

---

> > > ### Author Response · Authors · 2020-11-25
> > > **Response to Reviewer4**
> > >
> > > Thank you for your response and great suggestions.
> > >
> > > (About confusion of naming) \
> > > We used “representation reuse” and “representation change” at an algorithm-level, i.e., “representation reuse” implies none of the representation layers change significantly. The concept of “representation change” is for “*not* representation reuse,” which means at least one representation layer varies significantly. As you commented, we explained the adaptations of BOIL in a layer-wise manner as “BOIL enjoys representation reuse at the low- and mid-levels of the body and representation change in a high-level of the body.” (Third bullet in Section 4.1).
> > >
> > > To avoid such naming confusion, we will define different terms for each algorithm-level and layer-level. We will keep using "representation change" and "representation reuse" at the algorithm-level, but use “representation *layer* change” and “representation *layer* reuse” at the layer-level. Then, *if there is at least one "representation layer change" among the body layers, we can say that the model possesses "representation change" property.* It is also possible that the model has "representation change," but some layers reuse their parameters and features. For example, the 4conv network trained by BOIL has “representation layer reuse” at (conv1, conv2, and conv3), and “representation layer change” at conv4. Because there is one “representation layer change” at conv4, the model has “representation change” property. In particular, we will revise the “representation change” and “representation reuse," used for layer-wise analyses, to the “representation layer change” and “representation layer reuse.”
> > >
> > > (Ablation study 1) \
> > > Single layer training results are added in Appendix M. We conduct the experiment not only all the cases training a single representation layer, but also the cases training a single representation layer and the output layer. On both miniImageNet and Cars, it is observed that training a higher-level representation layer (conv3, conv4) without updating the head performs better than training any single conv layer with the output layer. We also observe that learning only a lower-level representation layer (conv1, conv2) can significantly decrease accuracy. The results reassure that representation layer change in higher-level layers of the body boosts the performance, as we noticed from the layer-wise analyses in section 4.
> > >
> > > We conduct additional experiments where we train not just a single layer but multiple consecutive layers. The results are also reported in Appendix M. From the results, we find that the straightforward BOIL algorithm has the near-best performance without a tremendous effort to find the best combination. We think that learning high-level representation layers and freezing the output layer are compatible and necessary to improve performance.
> > >
> > > (Ablation study 2) \
> > > The results are added to Appendix N. As you commented, BOIL, i.e., the head's learning rate of 0, performs the best. However, the accuracy rapidly decreases as the learning rate of the head grows. Even with 1/10X head learning rate compared to other layers, the test accuracies are significantly degraded. Based on this result, we think the freezing head is crucial.
> > >
> > > (Planning) \
> > > In summary, throughout additional experiments, it is confirmed that having the upper layer adapting without head leads to performance boosting. In order to prevent any confusion from names, we will revise the paper. And we are planning to add additional experiments with more deeper networks like more deeper ResNet with various datasets such as miniImagenet, tiredImagenet, Cars, CUB, Aircraft.

---

### Official Review · AnonReviewer1 · 2020-10-29
**An interesting exploration, but I have a concern about whether BOIL is really working through representation change.**

**Rating:** 7
**Confidence:** 4

**Review:**

This paper performs an investigation (with an associated new algorithm, BOIL), into the relative importance of representation change vs representation reuse in few-shot learning with MAML. It also compares to related prior work, 'Rapid Learning or Feature Reuse, (Raghu et al 2020)', that explored this question, finding representation reuse to be the key component of MAML, and their associated algorithm, ANIL, which only updates the network head (not the body) in the inner loop of training.

The paper studies the important role played by representation change when performing few-shot learning across domains, and to explore this, proposes an algorithm BOIL (Body Only in the Inner Loop), which, in contrast, only updates the network *body* in the inner loop, and keeps the network head frozen. The results show that BOIL matches performance of MAML/ANIL in single domain few-shot learning (mini ImageNet, tiered MiniImagenet), and provides performance improvements in fine-grained and cross-domain few shot learning (Cars, CUB, cross-domain between these datasets).

While the results are interesting, I have a key concern on how BOIL is working and some of the analysis results:

By freezing the head of the network during inner loop training, the head effectively acts as a fixed random projection layer, and  it is very possible that the penultimate layer now acts like a pseudo-head. This is what seems to be indicated by Figure 3, where the three first layers comprising of the body have extremely high representation reuse before/after adaptation, and the fourth layer, becomes much less similar, possibly acting as a head. It is important to determine whether this is what is happening, because it affects how much BOIL is actually reliant on representation change.

Before recommending accept, I would like to see more CKA comparisons (and any other representation analysis) of the BOIL representations across the different architectures/datasets, to confirm that BOIL is indeed working through representation change.

Minor Comments:
--- It would be helpful to know how the few shot learning tasks are setup on Cars and CUB
--- The key detail of which representations are being compared in Figure 2 (cosine similarity plot) is very unclear. Also don't understand why representations before/after adaption are being plotted separately, instead of cosine similarity applied to before/after pair.

--------------------------------------
--------------------------------------
--------------------------------------
**After Rebuttal and Discussion**
I appreciate the changes and additional experiments added by the authors, and am recommending accept.

For the final version, I would strongly encourage the authors to make the representation change/representation reuse argument clearer. Specifically, the authors might want to first present the BOIL algorithm, highlight key aspects of the algorithm (frozen head), present performance results, and then the ablation results in Appendix N to highlight the importance of freezing the head. After this, the paper could switch to a discussion on representation reuse/change and present the analysis results, making clear what is happening at a layer level vs at the algorithm level.

Right now, I think it's still a little confusing that representation change often refers to the algorithm not the layers, and this actually reduces the impact of what is a very striking result!

I hope the authors can make these changes, and with the clearer messaging, this should be a very interesting paper for the community, and provide many interesting directions for future work!

---

> ### Author Response · Authors · 2020-11-15
> **Replies for your valuable review**
>
> Thank you for carefully reviewing our paper and for the insightful and constructive comments. We have addressed your comments and updated our manuscript accordingly. Please find our detailed answers below.
>
> 1.	We do not believe that the penultimate layer acts like a pseudo-head. We believe that the role of the head layer is to draw a simple decision boundary for the high-level features represented by the output of the last CONV layer. However, the input of the penultimate layer cannot be simply classified (i.e., the desired performance cannot be achieved through the input of the penultimate layer) and the representation power of the input of the penultimate layer (i.e., representations after conv3) are similar for all MAML, ANIL, and BOIL (Appendix L, Table 16). Indeed, the penultimate layer acts as a non-linear transformation so that the fixed output head layer of BOIL can easily conduct a classifier role. We examined this claim with several different views. \
> \
> 1-1. Cosine similarity shown in Figs. 2 and 4, and Figs. 15 and 22 in the Appendix. The cosine similarity gap between the inter-class and intra-class in MAML, ANIL, and BOIL is always near zero before the penultimate layer (i.e., conv1-to-conv3 or block1-to-block3) before adaptation. The zero-gap means that the features cannot be simply classified. \
> \
> 1-2. NIL test results in Table 4, and Tables 13, 14, and 15 in the Appendix. The NIL test accuracy of MAML/ANIL is almost the same before and after the inner loop adaptation, which means they have well-represented features at the penultimate layer. By contrast, the NIL test accuracy of BOIL before adaptation is much worse than that after adaptation. From this, we believe BOIL makes well-trainable features at the penultimate layer, allowing the features to move quickly to the inside of the predefined decision boundary by the output head. \
> \
> 1-3. Norms of gradients in Fig. 17 in the Appendix. We plotted the average norm of the gradients of each layer. BOIL makes all conv layers have much larger gradient norms compared to MAML/ANIL. The gradient statistics support that MAML/ANIL simply reuses all conv layer features, whereas BOIL learns the features.
>
>
> 2.	We added additional cosine similarity, CKA, and empirical results (NIL-testing) to support representation change of BOIL. \
> \
> 2-1. Section 4 includes the above three factors (4conv network/miniImageNet); \
> \
> 2-2. Appendix H includes the above three factors (4conv network/Cars); \
> \
> 2-3. Section 5 and Appendix K include the above three factors (ResNet/miniImageNet and ResNet/Cars). \
> All cases show the same trend.
> In addition, Appendix I provides gradient norms according to the algorithm.
>
>
> 3.	(Minor comments) We think your first minor comment is related to the dataset split method, which is described in Appendix A.3. \
> In section 4.1, the details of Figure 2 are now given as follows: “We first investigate the cosine similarity between the representations of a query set including 5 classes and 15 samples per class from miniImageNet after every convolution module.” \
> We provide figures of cosine similarities for the before/after adaptation cases separately because the figures can describe how separable the representations are at each layer before/after adaptation. For instance, from the figures, we can demonstrate that the representations of conv4 with MAML/ANIL are already separable prior to adaptation.

---

> > ### Comment · AnonReviewer1 · 2020-11-24
> > **Response to authors (and in agreement with Reviewer4's comment)**
> >
> > Thank you to the authors for their detailed response and paper revisions, from which it is clear to me that the penultimate layer is performing some non-trivial function. However, then the accurate summary is exactly as reviewer4 has described in their comment, where the forced adaption of the penultimate layer is significantly helping performance.
> >
> > This is a very interesting result! But I also feel like framing it in terms of representation reuse and representation change is a little misleading, because all of the lower layers are clearly performing representation reuse.
> >
> > One experiment that might help with determining what is happening is also analyzing a deeper architecture. You have experiments on ResNet, but right now are only looking at activations at the end of each block. Would it be possible to plot results of all/a subset of activations in block3 and block4 across MAML/ANIL/BOIL to see if more layers than just the penultimate layer are showing representation change?
> >
> > To summarize: the fact that BOIL helps so much with performance is already very interesting, and I would just like to see a more coherent explanation of what is happening with the representations. I don't think whether the underlying effect is more reuse or change, or (as I suspect) a mix of both, takes anything away from the interesting aspects of the algorithm and its contribution, but it does feel important to articulate this correctly.
> >
> > (From the evidence you've presented, my hypothesis if you tried this out on a larger architecture is that you'd indeed see a mix of reuse and change.)
> >
> > Would you be able to reframe the paper appropriately and include additional experiments in the final revision? (Even if it doesn't happen by the end of the discussion period?)

---

> > > ### Author Response · Authors · 2020-11-25
> > > **Response to Reviewer1**
> > >
> > > Thank you for your thoughtful reading of our responses and other reviewers' comments.
> > >
> > > (About confusion of naming) \
> > > Thanks to reviewers’ comments, we identify the confusion in our representation change/reuse terms. We will clarify it. The terms at layer-level will be distinguished from algorithm-level representation change/reuse terms by using *representation "layer" change/reuse* to prevent confusion. As in answering to R4, if at least one layer is “representation layer change”, the model has “representation change” property at algorithm-level. On the contrary, if all layers are “representation layer reuse”, the representations do not significantly change through adaptation, so it is referred to as “representation reuse” at the algorithm-level.
> > >
> > > (Representation “layer” change / reuse in ResNet) \
> > > We add the plot of layer-wise cosine similarity of the last block of ResNet in MAML/ANIL and BOIL to check the representation change in a deeper network architecture. The results are in Appendix O.
> > >
> > > In MAML/ANIL, there is little or no representation layer change in all layers through adaptation. In contrast, in BOIL, the similarity gap on some layers between intra-class similarity and inter-class similarity is enlarged through adaptation, which indicates the representation change.
> > >
> > > Besides, as you commented, in BOIL, the “representation layer reuse” and “representation layer change” are mixed within the block. Furthermore,  it is observed that representation at high-level layers in the last block changes more when the last skip connection does not exist than when the last skip connection exists. These results confirm that the disconnection trick strengthens “representation layer change” at high-level layers by not directly propagating general representations from prior blocks.
> > >
> > > (Planning) \
> > > Because of the time limitation, we added only cosine similarities in the last block on miniImageNet and Cars. For the final revision, we will add CKA and cosine similarities plot and analyses on them in other blocks with other datasets. In particular, we will revise the paper to prevent any confusion from names.
> > >
> > > We also would like to point our new answer to Reviewer 4. We have tested additional experiments with interesting new settings that can explain the reasons for the superior performance of BOIL.

---

### Official Review · AnonReviewer3 · 2020-10-29
**A well written paper which might benefit from more discussion.**

**Rating:** 7
**Confidence:** 4

**Review:**

Summary:
Previous work studied MAML and showed that representation reuse is the main contributing factor in performance and not representation change. This paper first asks the question of whether representation reuse is enough for meta-learning? The paper hypothesizes and empirically evaluates the need/benefit for representation change in meta-learning tasks especially for cross-domain transfer. For this, they propose BOIL, a variant of MAML where the head of the network is not updated during inner loop updates.

Strengths:
1. The paper is very well written.
2. They do a simple, clearly described empirical study to understand what changes in representation happen in MAML during its training/testing and how to modify the MAML to encourage representation change.
3. The proposed change to MAML, BOIL shows improved performance in several of the benchmark datasets especially in cross-domain tasks (compared to MAML)
4. Their overall study is beneficial for the gradient based meta-learning models similar to MAML to better understand what is happening and directions to modification to training that could lead to potential improvement in generalization performance.

Weakness/Comments:
1. While the paper focuses on MAML, it is not clear what and how their findings and design modifications apply for other meta-learning methods such as CNP and also to gradient based meta-learning in Reinforcement Learning. Some discussion on the scope of the finidings/methods would be very helpful.
2. I am concerned if what is happening in BOIL is that the problem of not have representation change is just being passed on to the body. i.e the top layer of body is like the head which is reusing and the everything below is changing. i.e I am wondering for deep networks the problem is just passed on to the lower part and not really handled. At some point we probably want the layers to change less and be reused. Authors comments on this would be appreciated.
3. Even in a perfect representation change setting, there will be limitation of generalization to test time tasks depending on the distribution of the training tasks. One would hope that this covers a much larger set of domain transfers than with the representation reuse case. Some brief discussion on this would be good. i.e BOIL may not be able to transfer to any cross domain, if not, what class of domains can it successfully transfer to?
4. My guess is that the speed of adaptation might decrease in BOIL compared to MAML. This might not be a problem in SL, but might be of interest in RL. What do you think?
5. Do we lose anything by not allowing the head to change during inner loop training? Would it be possible to alternate between lr of inner loop being zero and non-zero for the head?
6. One can imagine if we had a lot of training tasks with a wide distribution and a model that cannot memorize all that the training of , and good optimizer, MAML would be forced to perform representation change as the training loss cannot be successfully reduced using just training reuse as the the model's capacity is limited. Would this be an alternative at the cost of more data? What are your thoughts?

Other Comments:
1. I think if we can achieve representation change, then it can also help make MAML less prone to memorization overfitting ("Meta-Learning without memorization, ICLR 2020, Meta-Learning requires meta-augmentation, NeurIPS 2020"). That might also a be something to explore.

Questions to authors:
Please reply to the above comments/weakness mentioned.

---

> ### Author Response · Authors · 2020-11-15
> **Replies for your valuable review**
>
> Thank you for your careful review of our paper and for the insightful and constructive comments. We have addressed your comments and updated our manuscript accordingly, particularly the Conclusion section. Please find our detailed answers to your questions.
>
> 1.	Because BOIL is also model-agnostic, we believe that our approach is also efficient when applied with other methods and in other fields. An example of this is given Section 3.2.2, in which BOIL is implemented to the WarpGrad meta-learning algorithm, improving the performance. We are also eager to apply BOIL to various other fields as our future research.
>
> 2.	We have added a plot of the average norm of the gradients for each layer. Please refer to Fig. 17 in Appendix I. This graph describes how much change has occurred in each layer. BOIL makes the penultimate layer adapt quickly while allowing other conv layers to reuse many parts.
>
> 3.	From our cross-domain results (Table 2 and 5), we found that BOIL overwhelms MAML/ANIL, particularly when the model is trained on a “fine-grained” dataset and is then evaluated on other datasets (e.g., Cars->miniImageNet or Cars->CUB). However, we believe that qualifying the distance between domains should be investigated to understand the utility of the representation change and reuse more clearly for the cross-domain setting. This will be another interesting future research topic.
>
> 4.	The adaptation speed of BOIL is faster than that of MAML/ANIL. We applied a single inner update for both meta-training and meta-testing with a large inner learning rate. Please refer to Section 2.2 for our experimental settings. Furthermore, BOIL also adapts rapidly with a small inner learning rate (Appendix E, Figure 12). We have added the results under the original MAML setting (5 inner steps for training, 10 inner steps for evaluation, and a 50-times smaller inner learning rate than our approach) in Appendix E. With the five inner steps setting, we observed that the performance of BOIL increases significantly with just one step. We believe that this characteristic is a promising signal for RL.
>
> 5.	We believe that, based on the numerous results of our study, not allowing the head to change during the inner loop is extremely close to the optimal. We also believe that, by updating the head, class-bias is injected into the model, harming the meta-initialization. Related to this issue, we discuss the ideal meta-initialization (please refer to Appendix J). The ideal meta-initialized head should not contain any prior information regarding the class indices.
>
> 6.	Thank you for your insightful comment. Our main idea from this study is that MAML/ANIL enforces a feature reuse and that such a reuse is not good for meta-learning. When there are too many tasks to memorize, the performance gap between BOIL and MAML/ANIL might decrease slightly.
>
> 7.	(Other comments) We agree that connecting a representation change to a memorization overfitting would be an interesting topic. We believe that the memorization overfitting problem of MAML might be related to task-specific (i.e., class-specific) bias, which is injected into the model (in particular, the output layer) despite the shuffling label (please refer to Answer #5). Therefore, BOIL might be less prone to memorization problem.

---

### Author Response · Authors · 2020-11-15
**We have uploaded a revised manuscript**

We really thank all the reviewers for their careful and valuable comments.

We have uploaded a revised draft considering your feedback and replied to each reviewer question by question.

---

### Author Response · Authors · 2020-11-25
**We have uploaded the second version of revision**

We thank all reviewers for their insightful and constructive comments.

Based on the comments from R1 and R4, we added three contents to the Appendix. \
Appendix M: ABLATION STUDY ON THE LEARNING LAYER \
Appendix N: ABLATION STUDY ON THE LEARNING RATE OF THE HEAD \
Appendix O: REPRESENTATION LAYER CHANGE IN THE LAST BLOCK OF RESNET-12

---

### Decision · Program_Chairs · 2021-01-07
**Final Decision**

**Decision:**

Accept (Poster)

**Comment:**

The paper presents a new algorithm, BOIL, on the importance of representation change vs reuse in MAML. All reviewers found the paper insightful, with some proposing a few changes to make the paper even stronger. Like them, I recommend accepting the paper.